# An optogenetic cell therapy to restore control of target muscles in an aggressive mouse model of amyotrophic lateral sclerosis

J Barney Bryson[1,2]*†, Alexandra Kourgiantaki[1,2], Dai Jiang[3], Andreas Demosthenous[3], Linda Greensmith[1,2]*†

[1]Department of Neuromuscular Diseases, UCL Queen Square Institute of Neurology, University College London, London, United Kingdom; [2]UCL Queen Square Motor Neuron Disease Centre, UCL Queen Square Institute of Neurology, University College London, London, United Kingdom; [3]Department of Electronic and Electrical Engineering, University College London, London, United Kingdom

**\*For correspondence:**
barney.bryson@ucl.ac.uk (JBB);
l.greensmith@ucl.ac.uk (LG)

†Co-Senior authors

**Competing interest:** The authors declare that no competing interests exist.

**Abstract** Breakdown of neuromuscular junctions (NMJs) is an early pathological hallmark of amyotrophic lateral sclerosis (ALS) that blocks neuromuscular transmission, leading to muscle weakness, paralysis and, ultimately, premature death. Currently, no therapies exist that can prevent progressive motor neuron degeneration, muscle denervation, or paralysis in ALS. Here, we report important advances in the development of an optogenetic, neural replacement strategy that can effectively restore innervation of severely affected skeletal muscles in the aggressive SOD1$^{G93A}$ mouse model of ALS, thus providing an interface to selectively control the function of targeted muscles using optical stimulation. We also identify a specific approach to confer complete survival of allogeneic replacement motor neurons. Furthermore, we demonstrate that an optical stimulation training paradigm can prevent atrophy of reinnervated muscle fibers and results in a tenfold increase in optically evoked contractile force. Together, these advances pave the way for an assistive therapy that could benefit all ALS patients.

## eLife assessment

This **fundamental** study presents a valuable method to restore muscle innervations in ALS mouse models using optogenetics. It is **convincing** that embryonic stem cell derived motor neurons can be transplanted into and applied to reinnervate the muscles in an ALS mouse model. The work will be of broad interest to researchers and medical biologists to develop new strategies for the treatment of neurodegenerative disorders resulting from denervated skeletal muscles.

## Introduction

The progressive degeneration of motor neurons that occurs in amyotrophic lateral sclerosis (ALS), the most common adult motor neuron disease (MND), affects almost all cellular components of the neuromuscular system, including cortical and spinal motor neurons, interneurons (*Crabé et al., 2020*), glial cells (*Van Harten et al., 2021*; *Vahsen et al., 2021*), as well as muscle (*Loeffler et al., 2016*). Motor axons, contained within peripheral nerves, serve as the final common relay for transmission of motor signals that control all voluntary muscle contraction and movement. However, one of the earliest characteristic pathological features of ALS involves 'die-back' of motor axon terminals (*Fischer et al.,*

*2004*) and breakdown of neuromuscular junctions (NMJs), the specialized synapses between motor axon terminals and muscle fibres. Effectively, this represents a single point of failure that permanently blocks motor signal transmission, irrespective of the condition of central motor circuits involved in coordination of movement signals. This results in an inexorable progression of muscle weakness, atrophy and, ultimately, complete paralysis, eventually leading to premature death. The median survival time in ALS, from initial onset of symptoms to death, typically as a result of respiratory complications, is only 20–48 months (*Chiò et al., 2009*) and ALS has an estimated global mortality of 30,000 patients per year (*Mathis et al., 2019*).

ALS is typically classified into either familial (fALS) or sporadic (sALS) forms of the disease, based on whether or not patients have an identified family history of the disease; between 5% and 10% of total ALS cases fall into the former category, fALS, with the remaining 90–95% consisting of sALS cases (*Mathis et al., 2019*). To date, over 20 monogenic mutations that cause ALS have been identified; however, these still only account for 45% of fALS cases and only 7% of sALS cases (*Mejzini et al., 2019*). The downstream cellular and molecular pathomechanisms underlying neurodegeneration in ALS are extremely complex and include dysregulation of proteostasis, autophagy, RNA metabolism and axon transport, as well as excitotoxicity, oxidative stress and neuroinflammation (*Mejzini et al., 2019*). Given the disparate causes and complex disease mechanisms, development of an effective therapy has proven extremely challenging and there are currently no effective treatments capable of arresting the progressive paralysis that occurs in ALS (*Brown and Al-Chalabi, 2017*). Even emergent gene therapy approaches, such as antisense oligonucleotides (ASOs) that have shown early promise in clinical trials (*Miller et al., 2013*), are unlikely to benefit the majority of patients with sporadic ALS and will not replace degenerated motor neurons or restore motor function once it has been lost.

Similarly, efforts to develop cell therapy approaches for ALS have, so far, primarily been aimed at slowing the degeneration of motor neurons in the spinal cord by providing neurotrophic factor (NTF) support through intraspinal engraftment of foetally derived neural stem cells (NSCs; *Feldman et al., 2014*; *Goutman et al., 2018*) or intrathecal delivery of autologous mesenchymal stem cells (MSCs) that are modified to overexpress NTFs (*Cudkowicz et al., 2022*; *Berry et al., 2019*). Whilst these approaches were shown to significantly slow disease progression in transgenic animal models of ALS (*Xu et al., 2006*; *Yan et al., 2006*), clinical trials in ALS patients have shown only a modest therapeutic effect in the case of intraspinal NSC grafts (*Glass et al., 2012*) and no overall benefit of intrathecal MSC delivery (*Cudkowicz et al., 2022*). Given the highly invasive surgical laminectomy required to engraft NSCs into the ventral horn of the spinal cord, only localized populations of motor neurons in the lumbar and/or cervical region were targeted; however, the surgical procedure was generally well tolerated and the approach was proven to be safe in clinical trials (*Feldman et al., 2014*; *Glass et al., 2016*). Moreover, these clinical trials have provided evidence that specific muscle functions can be preserved for longer in some ALS patients (*Mazzini et al., 2019*). Importantly, this pioneering approach provides a precedent for implementation of an allogeneic stem cell-based therapy and also shows that ALS patients can tolerate a 6-month period of immunosuppression (*Mazzini et al., 2015*), which appears to be sufficient to confer long-term survival of engrafted cells (*Tadesse et al., 2014*). Nonetheless, it appears unlikely that this approach will be able to restore motor function once it has been lost, since the NSCs do not replace lost motor neurons, and any therapeutic effect has so far been shown to be transient (*Goutman et al., 2018*; *Mazzini et al., 2019*). Therefore, there remains an urgent unmet need to develop novel therapies that can rescue muscle innervation and maintain muscle function in ALS patients.

We have previously demonstrated a novel proof-of-concept strategy to overcome muscle denervation and restore control of muscle contraction in a nerve injury model of muscle paralysis that could have major therapeutic value for restoring function of any targeted muscle or group of muscles in ALS patients (*Bryson et al., 2014*). Briefly, optogenetically modified replacement motor neurons, derived from murine embryonic stem cells (mESCs), were engrafted into distal branches of peripheral nerves supplying specific lower hindlimb flexor and extensor target muscles in wildtype mice that had undergone a nerve ligation injury. Our results showed that the engrafted motor neurons were able to project axons from the graft site to the target muscles where they formed de novo NMJs. Due to the ectopic location of the engrafted motor neurons, outside the CNS, they do not receive endogenous motor signals, and must therefore be exogenously activated. Expression of the blue-light sensitive channelrhodopsin-2 (ChR2) protein (*Nagel et al., 2003*) in the engrafted motor neurons conferred

the ability to selectively activate these engrafted neurons and thereby control the contractile function of the target muscle using acute optical stimulation (*Bryson et al., 2014*). The aim of this neural replacement strategy is therefore to provide a biological interface capable of rendering any target muscle receptive to control signals transmitted by optical stimulation to engrafted motor neurons (*Bryson et al., 2016*). Importantly, we have recently developed a prototype 64-channel stimulation and recording device capable of controlling multiple independent intraneural graft sites that could be used to elicit coordinated function of large numbers of muscles, in order to restore useful motor functions (*Liu et al., 2022*).

This novel approach to restore control of paralyzed muscles in ALS patients, using a combination of cell replacement and optical stimulation, has several key advantages over existing cell replacement and electrical stimulation strategies, including: (i) the ability to engraft motor neurons peripherally, in close proximity to targeted muscles, which greatly accelerates the rate of reinnervation and reduces the period of denervation, consequently ameliorating denervation-induced muscle atrophy; (ii) avoidance of engrafting replacement cells into the neurotoxic environment that exists within the CNS of ALS patients and the necessity for reinnervating axons of CNS-engrafted motor neurons to overcome the inhibitory CNS:PNS barrier in order to exit the CNS and grow the often long distances to target muscles; (iii) specificity of optical stimulation to the engrafted ChR2$^+$ motor neurons avoids painful off-target activation of sensory afferents or aberrant activation of endogenous motor axons associated with electrical nerve stimulation (ENS); and critically, (iv) the ability to recruit motor units in correct physiological order using optical nerve stimulation (ONS; *Llewellyn et al., 2010*) avoids the problem of rapid muscle fatigue associated with ENS and incorrect, non-physiological motor unit recruitment. Furthermore, ENS-mediated control of muscle function depends on the presence of surviving motor axons and, since these are progressively lost during the course of disease progression in ALS, the ability of ENS to induce muscle contraction is steadily eroded. More importantly, it has recently been shown that ENS, applied to the phrenic nerve to assist respiratory function in two separate clinical trials in ALS patients (*McDermott et al., 2016*; *Gonzalez-Bermejo et al., 2016*), accelerated diaphragm muscle denervation, which likely contributed to a significant reduction in patient life-span (*Guimarães-Costa et al., 2019*). Therefore, it is unlikely that ENS will ever be suitable for artificial control of critical muscle function in ALS patients. In contrast, optogenetic stimulation (*Henderson et al., 2009*), in combination with neural replacement, represents a safe alternative approach to artificially restore innervation and function of paralyzed muscles in ALS.

In the current study, we sought to optimize critical elements of this novel therapeutic strategy and to determine whether ChR2$^+$ motor neurons can be used to successfully restore innervation, induce muscle contraction and prevent atrophy of targeted muscles in the highly aggressive SOD1$^{G93A}$ mouse model of ALS.

## Results

### Host-vs-graft rejection causes loss of most intraneural ESC-MNs allografts

ChR2$^+$ motor neurons were derived from our previously characterized mouse embryonic stem cell (mESC) line (*Bryson et al., 2014*) and differentiated using a well-established protocol (*Wichterle et al., 2002*). Since these donor ChR2$^+$ motor neurons were generated from an mESC line originating from the 129S1/SvImJ mouse strain (*Appendix 1—table 1*) and recipient mice SOD1$^{G93A}$ mice were on a congenic C57BL/6 J genetic background, they constitute an allogeneic source of donor cells. Importantly, allogeneic cells are likely to provide a more cost-effective, off-the-shelf cell therapy platform, compared to generating individual, patient-specific batches of cells suitable for autologous engraftment. Therefore, an important initial objective of this study was to identify an immunosuppression regimen capable of preventing host-vs-graft rejection of allogeneic ChR2$^+$ motor neurons to enable their innervation of target muscles in recipient SOD1$^{G93A}$ mice.

In the absence of any immunosuppression, the survival rate of ChR2$^+$ motor neuron grafts at 35 d post engraftment was extremely low (<5%) in SOD1$^{G93A}$ mice. However, in the rare cases where the engrafted ChR2$^+$ motor neurons did survive, we observed robust intramuscular axon growth and partial reinnervation of NMJs, even at very late-stage disease (*Figure 1A–C* and *Video 1*). Furthermore, acute optical stimulation of the engrafted ChR2$^+$ motor neurons was able to induce tetanic

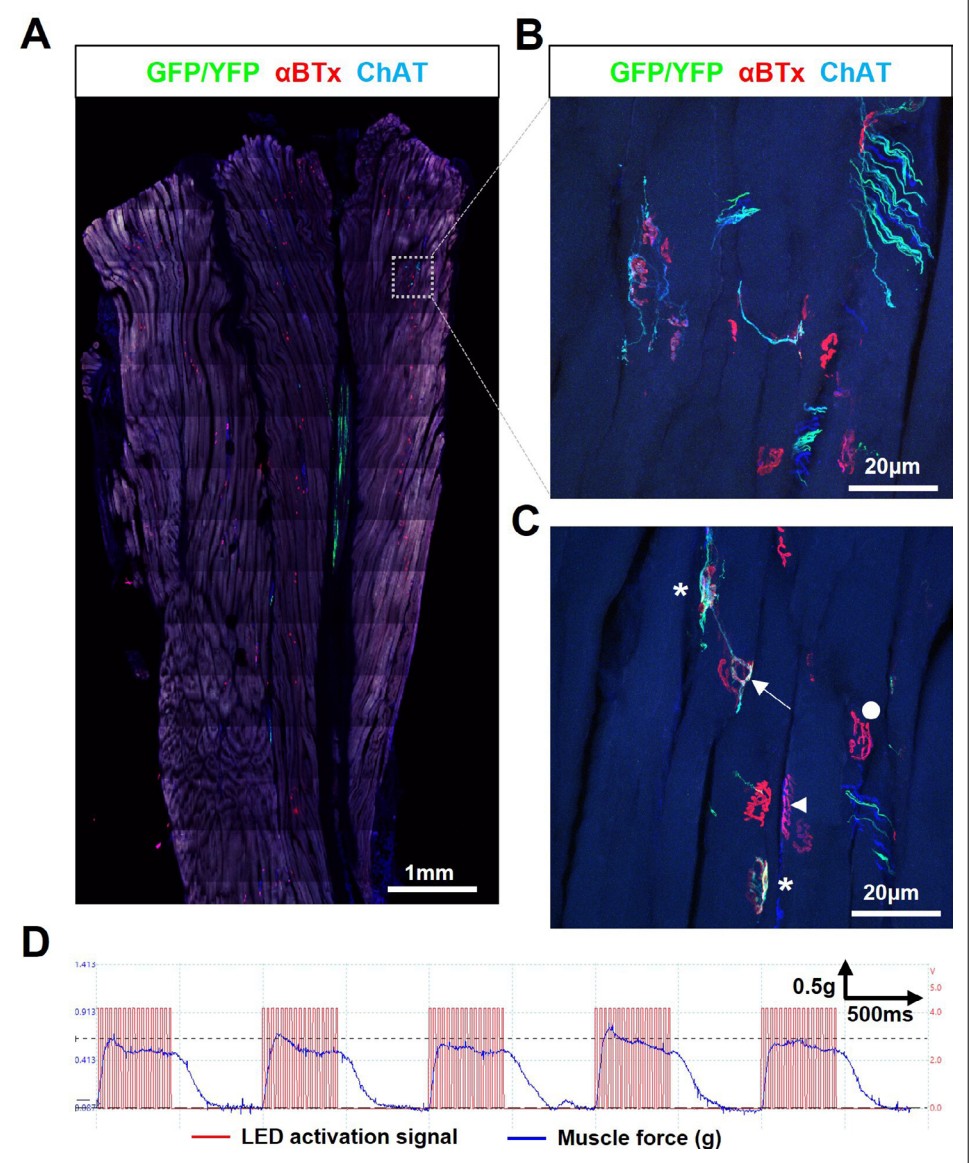

**Figure 1.** Engrafted allogeneic ChR2[+] motor neuron can survive and robustly reinnervate target muscles in SOD1[G93A] mice but occurs rarely. (**A**) Confocal tile-scan of a longitudinal section of the triceps surae muscle from a 135d SOD1[G93A] mouse, 45d post-engraftment, showing a rare example of graft survival, in the absence of immunosuppression, with robust intramuscular axon projection. (**B**) Maximum intensity projection (MIP) images of a confocal z-stack through region of interest (dashed box in (**A**)) and (**C**), showing NMJs fully (asterisks) and partially (arrow) innervated by ChR2[+] motor neuron axons, as well as innervated by endogenous (GFP/YFP-negative, ChAT-positive) motor axons (arrowhead) and fully denervated endplates (circle). (**D**) Representative recording (n=1/3 positive responders) showing characteristically weak contractile responses to repetitive 20 Hz optical stimulation.

contraction of the target muscle in these SOD1[G93A] mice (*Figure 1D*), although these were weak in magnitude. Interestingly, assessment of shorter timepoints revealed that engrafted ChR2[+] motor neurons were capable of surviving for up to 14 days (data not shown). It is therefore likely that the poor long-term survival was due to host-vs-graft rejection of the engrafted cells, rather than disease-related toxicity. These findings suggest that if the engrafted motor neurons can evade the host immune response, they can functionally reinnervate target muscles for a therapeutically relevant timescale in this highly aggressive model of ALS.

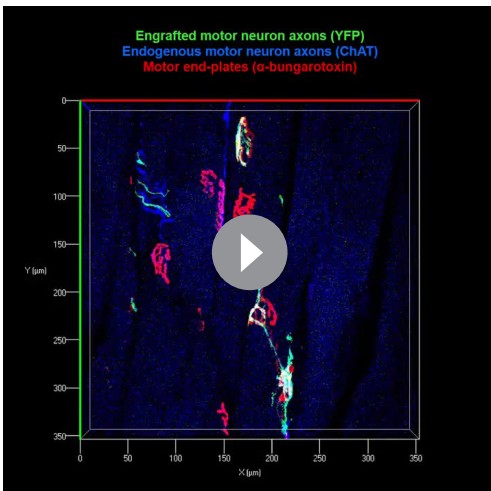

**Video 1.** 3D reconstruction of innervated endplates from SOD1^G93A mice in the absence of immunosuppression.

https://elifesciences.org/articles/88250/figures#video1

## Tacrolimus (FK506) overcomes ESC-MN allograft rejection but inhibits muscle innervation

In an effort to enhance graft survival, we first tested the calcineurin inhibitor (CNI), tacrolimus (FK506), which is not only routinely used to facilitate solid organ allograft survival in humans but has also been reported to promote axon regeneration after nerve injury in rats (*Gold et al., 1995*). Cohorts of wild-type and SOD1$^{G93A}$ mice (*Appendix 1—table 2*) were treated with FK506 (5 mg, kg$^{-1}$, d$^{-1}$; dose selected based on evidence of axon regeneration studies) from the time of intraneural ChR2$^+$ motor neuron engraftment; graft survival and muscle reinnervation were then assessed 30-35d post-engraftment. Although FK506 did enable robust ChR2$^+$ motor neuron allograft survival, at the graft site, in all animals examined (*Appendix 1—table 2*), we identified three major problems with this immunosuppressant agent. Firstly, although the engrafted motor neurons were able to survive and project axons along peripheral nerve branches and within the targeted lower hindlimb muscles (*Figure 2A*), FK506 completely prevented muscle fibre reinnervation, evidenced by lack of response to acute ONS (data not shown) and complete absence of ChR2$^+$/YFP$^+$ NMJs (*Figure 2B*) in both SOD1$^{G93A}$ (n=8) and nerve-ligated wild-type mice (n=4). Secondly, exuberant growth of carry-over pluripotent stem cells led to intraneural tumour formation in most mice (*Figure 2C* and *Appendix 1—table 2*), which resulted in overt hindlimb motor deficits (*Figure 2D* and *Video 2*). Thirdly, in partial agreement with previous reports in rats (*Aydin et al., 2004*), FK506 prevented body mass increase and/or induced body mass decline in a subset (44.4%; n=8/18) of SOD1$^{G93A}$ mice (*Figure 2E* and *Figure 2—figure supplement 1A*); this effect was less evident in wild-type mice (*Figure 2—figure supplement 1B*). Onset of body mass decline in B6.SOD1$^{G93A}$ mice is highly consistent and typically occurs at 115 days (*Hayworth and Gonzalez-Lima, 2009*), indicating that, at this relatively high dose, FK506 may be preferentially toxic or may exacerbate disease phenotype in SOD1$^{G93A}$ mice.

Since peripheral neuropathy is a known adverse event associated with calcineurin inhibitors (*Arnold et al., 2013*), it is possible that FK506 treatment alone may adversely affect endogenous or engrafted motor axons. Indeed, examination of the cross-sectional area distribution of total (i.e. sensory and motor) and motor neuron axons in branches of the sciatic nerve in FK506-treated SOD1$^{G93A}$ and WT mice (*Figure 3A*) revealed a significant loss of axons that affected most axonal calibres in the tibial nerve in wild-type and SOD1$^{G93A}$ mice (*Figure 3B and C*). A more pronounced loss of total and motor axons, spanning medium to large sized axonal calibres, was observed in the common peroneal nerve branch (*Figure 3D and E*), which indicates that FK506 can not only exacerbate ongoing motor axon loss in SOD1$^{G93A}$ mice but can also induce motor axon loss even in wild-type mice. Importantly, these neuropathy-like effects appear to be specific to FK-506, since an alternative immunosuppressant, H57-597 mAb (discussed in detail below), did not significantly alter total or motor axon size distribution or total axon counts (*Figure 3—figure supplement 1*), compared to untreated SOD1$^{G93A}$ mice.

Since FK506 is known to suppress myoblast proliferation and differentiation (*Hong et al., 2002*) and can cause rare cases of myopathy in humans (*Breil and Chariot, 1999*), it is also possible that the FK506-dependent prevention of muscle reinnervation by engrafted ChR2$^+$ motor neurons is due to a muscle specific effect. In any case, these findings clearly show that FK506 is unsuitable as an immunosuppressant to support ChR2$^+$ motor neuron allograft survival and, indeed, suggest that long-term administration at this relatively high dose should be avoided in ALS patients. Importantly, however, the complete protection of engrafted ChR2$^+$ motor neurons by FK506 did confirm that host-vs-graft rejection was responsible for the poor long-term graft survival observed in the absence of immunosuppression, rather than disease related neurotoxicity.

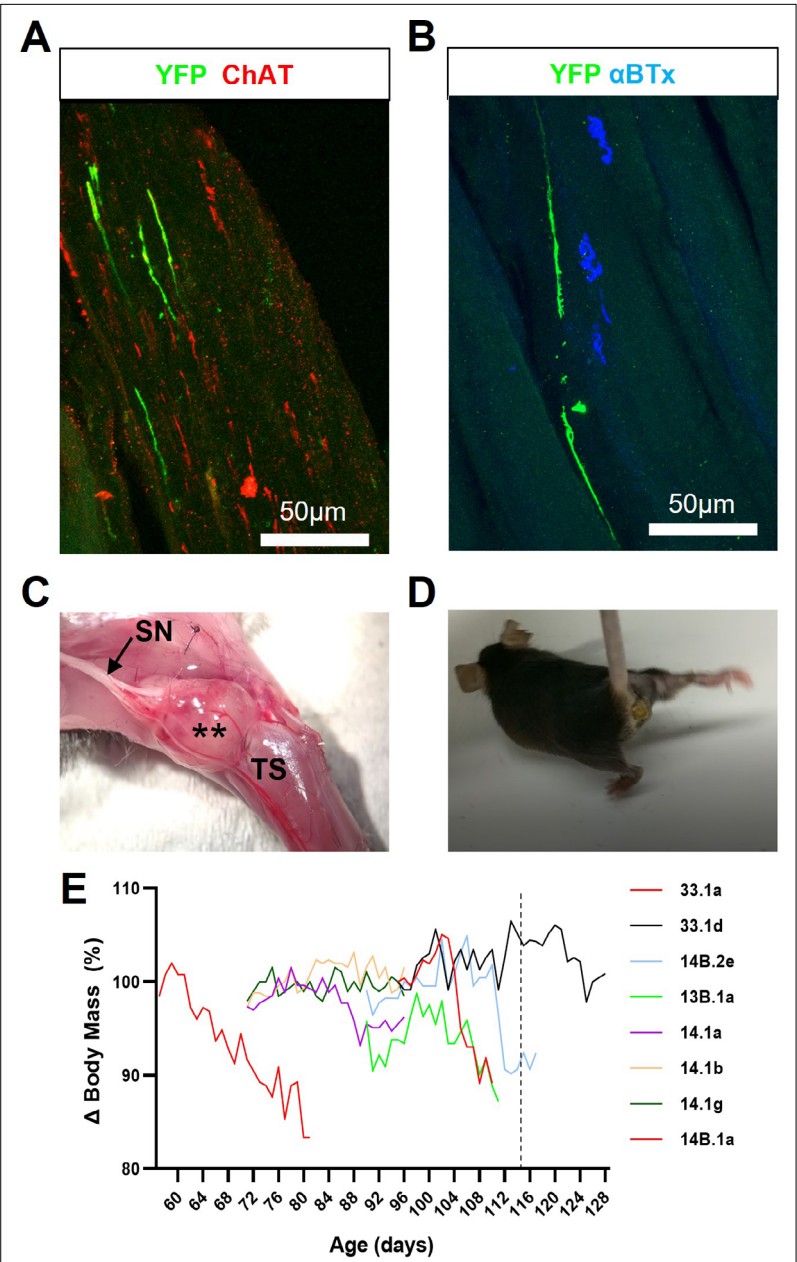

**Figure 2.** FK506 enables long-term survival of engrafted ChR2+ motor neurons but inhibits muscle reinnervation and exacerbates disease progression in SOD1$^{G93A}$ mice. (**A**) Representative confocal image showing that GFP/YFP+ axons are able to project within intramuscular branches, following intraneural engraftment of ChR2+ motor neurons and daily immunosuppression with FK506; obtained from a 112d SOD1$^{G93A}$ mouse at 27d post-engraftment. (**B**) GFP/YFP+ axons fail to reinnervate NMJs despite the proximity of ChR2+ motor axon terminals to denervated endplates. (**C**) FK506-mediated immunosuppression permits exuberant growth of carry-over pluripotent stem cells that form intraneural tumours (**) within engrafted sciatic nerve (SN) branches; (**D**) these tumours cause severe movement impairment of the affected hindlimb. (**E**) FK506 caused body mass loss in a subset (8/18) of SOD1$^{G93A}$ mice, treated at different ages, that precedes onset of normal decline in this model (indicated by vertical dashed line at 115d).

The online version of this article includes the following figure supplement(s) for figure 2:

**Figure supplement 1.** FK506 severely affects body mass in most SOD1G93A but not wild-type mice.

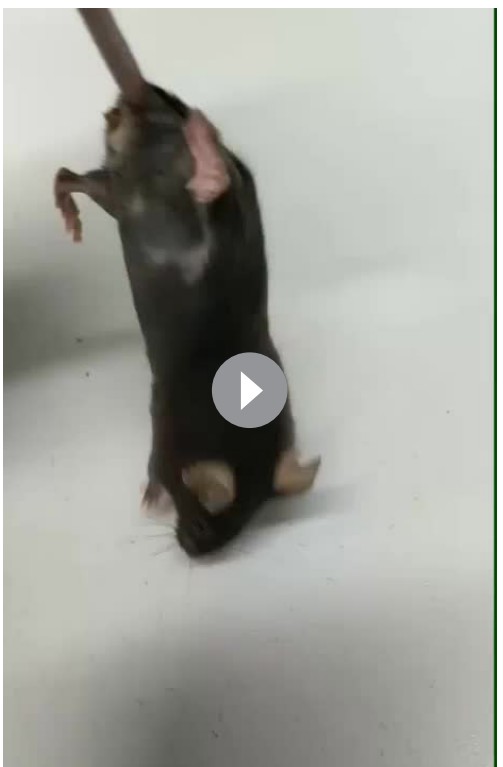

**Video 2.** FK506 facilitates graft survival but allows intraneural tumour formation that causes severe motor dysfunction.

https://elifesciences.org/articles/88250/figures#video2

## T-cell modulatory immunosuppression confers graft survival and target muscle innervation

In light of the deleterious effects of FK506, and given our aim of conferring compatibility of allogeneic ChR2+ motor neurons as a universal cell therapy for ALS, we sought to identify a more specific form of immunosuppression that avoids the negative effects of FK506 yet supports long term graft survival. Therefore, we investigated the T-cell receptor-β (TCR-β) targeting monoclonal antibody, mAb H57-597, which has previously been shown to effectively promote long-term heart allograft survival in mice (*Miyahara et al., 2012*). In addition, since our findings indicated that immunosuppression results in a greater risk of tumour formation from carry-over pluripotent stem cells, differentiated ChR2+ motor neurons were also pre-treated with mitomycin-C (MMC; 2 µg/ml for 2 hr) prior to engraftment, to eliminate tumorigenic cells (*Magown et al., 2016*), to further enhance the translational potential. MMC-treated ChR2+ motor neurons were unilaterally engrafted into the tibial nerve of symptomatic SOD1^G93A mice (aged 95.7±4.6 days) in conjunction with transient H57-597 mAb delivery (1 mg, kg-1; i.p. injection at 0, 1-, 3-, 7-, and 14 days post-engraftment). The extent of reinnervation and the ability to optically control the function of the triceps surae (TS) muscle group in the lower hindlimb of SOD1^G93A mice was then assessed at late-stage disease (133±6.9 d; n=12) by determining the physiological response of the reinnervated muscles to acute ONS of the engrafted motor neurons followed by histological analysis of muscle and nerve. Histological analysis confirmed that in mAb H57-597 treated animals, engrafted ChR2+ motor neurons were present in all recipient SOD1^G93A mice (n>84 to date). Importantly, there was significant axonal projection within intramuscular nerve branches and robust reinnervation of muscle fibres in the targeted TS muscle (*Figure 4A, B*; *Videos 3 and 4*). As we previously reported in a wild-type mouse nerve ligation model *Bryson et al., 2014*, some de novo NMJs in SOD1^G93A mice exhibited signs of immaturity, including poly-innervation (*Figure 4C*), as well as collateral and terminal sprouting of motor axons (*Figure 4D*). Since the peripherally-engrafted reinnervating motor neurons are inactive during the post-engraftment period and progressive muscle atrophy is ongoing, only 10.5% of endplates are innervated by engrafted ChR2+ motor neurons (*Figure 4E*). Importantly, these reinnervated endplates are functional, since acute in vivo ONS of the engrafted ChR2+ motor neurons in the exposed sciatic nerve of late-stage disease SOD1^G93A mice at 133±7.2 days of age (37.7±5.1 days post-engraftment; n=12), induced positive contractile responses in all animals, although the amplitude of the maximal contractile force was still weak (0.8±0.2 g; n=11).

## Motor neuron subtype identity does not affect response to optical stimulation

In an effort to increase the amplitude of the contractile response of the target muscle to optical stimulation, we next tested whether engraftment of motor neurons with a fast-firing subtype identity may be more suitable than engraftment of predominantly slow-firing motor neurons by using alternative differentiation protocols (*Wichterle et al., 2002*; *Peljto et al., 2010*). The stronger, predominantly fast-twitch, gastrocnemius component of the TS muscle group is usually innervated by fast-firing/

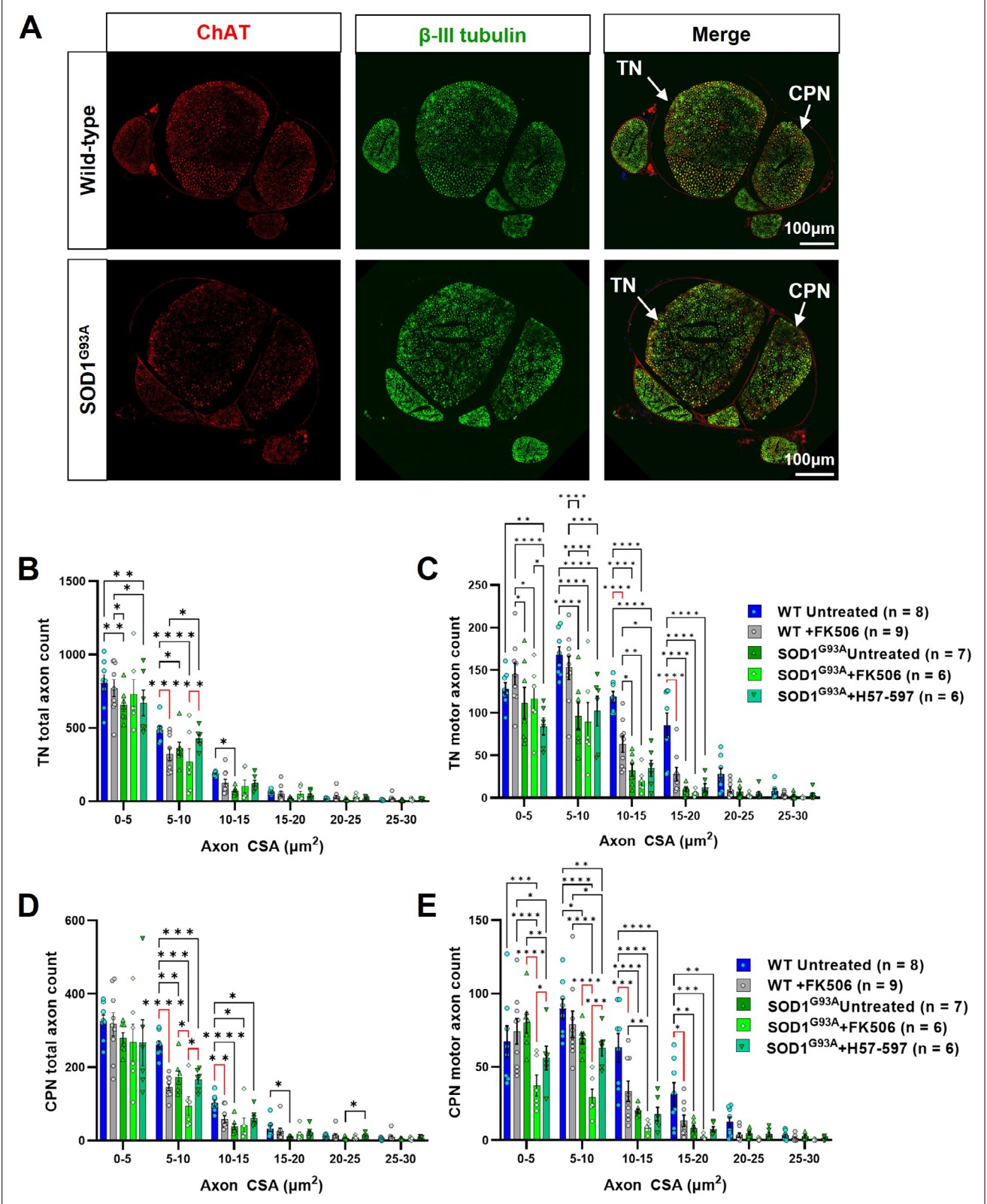

**Figure 3.** FK506 causes peripheral nerve axonopathy in SOD1G93A and wild-type mice. (**A**) Representative examples of wild-type (top) and SOD1G93A (bottom) sciatic nerve transverse sections, showing common peroneal nerve (CPN) and tibial nerve (TN), labeled for total axons (βIII tubulin; green) and motor axons (choline acetyl transferase; ChAT; red); automated axon size distribution analysis of (**B**) total and (**C**) motor axon in the tibial nerve (TN); (**D**) total and (**E**) motor axons in the common peroneal nerve (CPN) in both wild-type and SOD1G93A mice. Data shown as mean; error bars = SEM;

*Figure 3 continued on next page*

*Figure 3 continued*

two-way ANOVA analysis: *denotes p ≤ 0.05; ** denotes p ≤ 0.0002; *** denotes p ≤0.002; **** denotes p ≤ 0.00002; significance bars displayed in red highlight changes directly attributable to FK506, independent of genotype.

The online version of this article includes the following figure supplement(s) for figure 3:

**Figure supplement 1.** FK506 moderately reduces total sciatic nerve axon counts in wild-type mice but loss of total and motor axons is not observed in SOD1[G93A] mice when all axon calibers are grouped.

fast-fatigable (FF) motor neurons, which are known to have the capacity to innervate many more individual muscle fibres per motor unit than slow-firing motor neurons, which normally innervate a much smaller number of weaker, slow-twitch muscle fibres, predominantly within the soleus and plantaris regions of the triceps surae. MMC-treated ChR2[+] motor neurons, differentiated to yield FF subtype identity motor neurons, were engrafted into the tibial nerve of (106±7.2 days) SOD1[G93A] mice, in combination with transient H57-597 mAb treatment. Maximum isometric muscle contraction of the TS muscle in response to acute optical stimuli was then determined at the same age (133.9±7.2 days, n=13) as previous grafts of predominantly slow-firing motor neurons (133±6.9 days, n=11). This physiological analysis revealed that the motor neuron subtype identity did not significantly affect amplitude of the muscle response to acute optical stimulation (*Figure 4—figure supplement 1A* and *Appendix 1—table 3*) and that the maximum contractile force elicited by ONS remained extremely weak compared to supra-maximal ENS (*Figure 4—figure supplement 1B* and *Appendix 1—table 3*). This result implies that, unlike during normal neuromuscular development, motor neuron subtype identity is not an important determinant of muscle fibre innervation in the mixed fibre type triceps surae muscle. This finding has significance for future clinical translation, since only a single subtype of motor neuron may be required to innervate a variety of different muscles. Only MMC-treated motor neurons with a slow-firing medial motor column identity were used in the subsequent experiments reported here. Since modification of motor neuron subtype identity did not enhance contractile response, our next challenge was to identify an effective method to enhance reinnervation and force generating capacity of the targeted muscle in response to optical stimulation.

## Optical stimulation training significantly enhances target muscle force generation

Spinal motor circuit development (*Milner and Landmesser, 1999*; *Li et al., 2005*) and NMJ formation/maintenance (*Sanes and Lichtman, 1999*) are known to be activity-dependent processes, thus, without regular stimulation, although peripherally engrafted ChR2[+] motor neurons may survive, they are unlikely to form mature NMJs and will therefore have little effect on declining muscle function and atrophy in SOD1[G93A] mice. Moreover, paralysis and atrophy of affected muscles proceed unchecked in the SOD1[G93A] mouse model of ALS. We therefore investigated whether regular activation of the engrafted ChR2[+] motor neurons, in conjunction with H57-597 mAb treatment, could enhance NMJ maturation and maximize the ability of target muscles to generate contractile force in response to acute optical stimulation in late-stage SOD1[G93A] mice. To do this, we adapted a wireless, fully implantable optical stimulation system (*Montgomery et al., 2015*), in order to implement a daily optical stimulation training regimen for engrafted mice. First, we modified the printed circuit board (PCB) design to simplify assembly (*Figure 5—figure supplement 1A*), implemented a new encapsulation method to ensure long-term survival of the devices after implantation (*Figure 5—figure supplement 1B*) and incorporated an RF signal switch and pulse controller to deliver precisely timed RF pulses to power a 470 nm light emitting diode (LED) connected to the implantable device (*Figure 5—figure supplement 1C*).

The modified optical stimulation devices were then surgically implanted in SOD1[G93A] mice concomitantly with intraneural engraftment of ChR2[+] motor neurons, with the trailing LED positioned in close apposition to the graft site (*Figure 5A*). Commencing at 14 d post-engraftment, to allow growing ChR2[+] motor axons sufficient time to reach the target muscle, the mice were transferred to a custom built chamber located above a resonance-frequency induction cavity for 1 hr per day, in order to undertake optical stimulation training (OST; *Figure 5B* and *Video 5*), using a bespoke pulse pattern that was empirically determined to elicit maximum contraction (*Figure 5—figure supplements 2–4*), followed by a 2 s rest interval. Following daily OST in engrafted SOD1[G93A] mice for 21 days, isometric

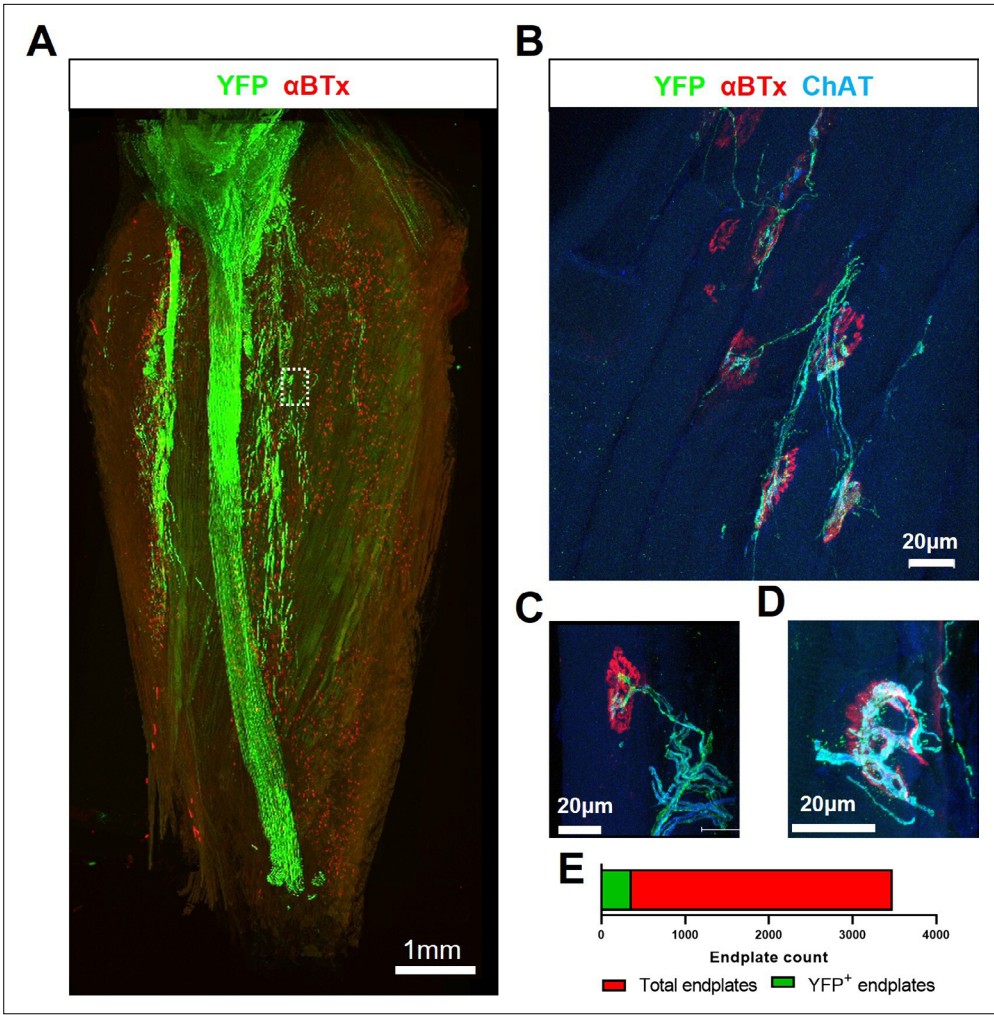

**Figure 4.** Transient H57-597 mAb treatment confers complete ChR2+ motor neuron allograft survival and, importantly, allows robust triceps surae muscle reinnervation up until late-stage disease in SOD1[G93A] mice. (**A**) 3D reconstruction of 67 individual tile-scans, acquired from serial sections from an entire triceps surae muscle, from a 135d SOD1[G93A] mouse following engraftment of ChR2+ motor neurons at 95d and H57-597 treatment, showing the full extent of axonal projection throughout the whole muscle; see also *Video 1*. (**B**) A high-resolution maximum intensity projection (MIP) image of a confocal z-stack, revealing multiple NMJs innervated to varying extents (α-bungarotoxin; α-BTx; red) by YFP + engrafted motor neuron axons (green) labeled for choline-acetyl transferase (ChAT; blue – note; overexposure of blue channel enables visualization of muscle fibres). (**C**) MIP image of a confocal z-stack showing an example of a partially innervated NMJ. (**D**) MIP image of a confocal z-stack showing an example of a fully innervated NMJ; note, poly-innervation, shown in (**C**), and terminal sprouting, shown in (**D**) which are signs of immaturity. (**E**) Automated quantification of total endplate number (count = 3482; labeled with α-BTx) and manual counts of endplates (count = 364) that showed YFP colocalization, indicating innervation, from the same muscle.

The online version of this article includes the following figure supplement(s) for figure 4:

**Figure supplement 1.** Subtype identity of engrafted ChR2+ motor neurons does not affect the maximum contractile response of the targeted muscle to acute optical stimulation in SOD1[G93A] mice.

---

muscle tension physiology was performed at late-stage disease to determine the maximal contractile force of the TS muscle elicited by acute ONS of the exposed sciatic nerve graft site. In confirmation of our hypothesis, there was a highly significant, 9.4-fold, increase in the maximal tetanic force (7.5±0.94 g versus 0.8±0.2 g; p=≤0.000001) elicited by ONS in the engrafted OST group of SOD1[G93A] mice at late-stage disease, compared to age-matched untrained SOD1[G93A] mice (132.4±6.8 days versus 133±6.9 days; n=7 and 11, respectively; *Figure 5C–G* and *Appendix 1—table 3*), although

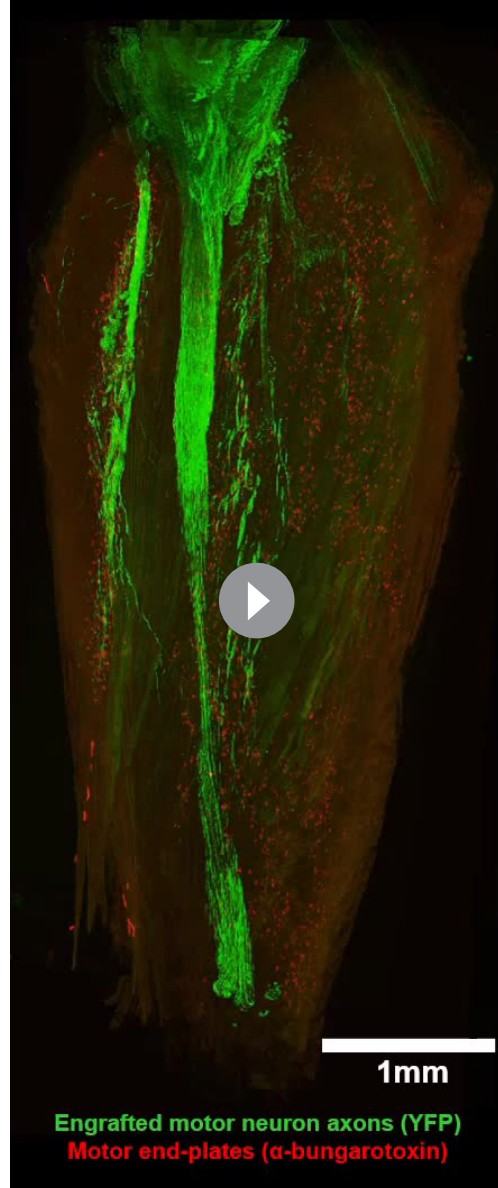

**Video 3.** 3D reconstruction of an entire triceps surae muscle group from a late-stage SOD1$^{G93A}$ mouse, after ChR2$^+$ motor neuron engraftment showing extent of reinnervation.
https://elifesciences.org/articles/88250/figures#video3

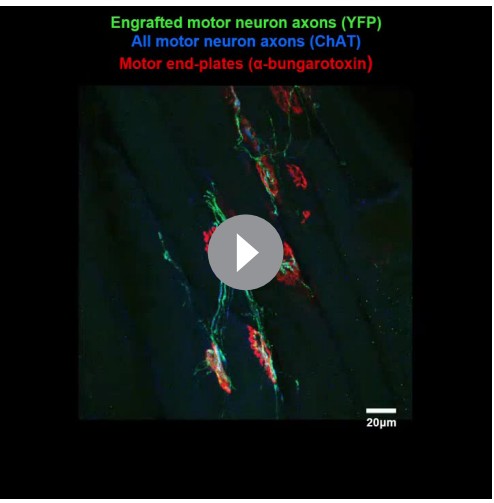

**Video 4.** 3D reconstruction of individual endplates (red) reinnervated by engrafted ChR2$^+$ motor neuron (green) in a 135d SOD1$^{G93A}$ mouse (35d post-engraftment) in combination with transient H57-597 mAb treatment.
https://elifesciences.org/articles/88250/figures#video4

OST did not alter contractile rate characteristics (*Figure 5—figure supplement 5A–C*). Moreover, quantitative comparison of the maximum force elicited by ONS compared to ENS of TS muscles in late-stage SOD1$^{G93A}$ mice (*Figure 5H*), showed that in mice that underwent OST, acute ONS elicits up to 22.7% (±4.5) of total residual muscle force produced by supra-maximal ENS, which activates both endogenous and engrafted motor neurons (*Figure 5I*), in contrast to only 1.46% (±0.18) in untrained SOD1$^{G93A}$ mice (*Figure 5—figure supplement 6A, B*). This represents a remarkable >13-fold improvement in force generation. In engrafted SOD1$^{G93A}$ mice that did not undergo OST, the extremely weak twitch contractions in response to ONS precluded interrogation of individual motor unit force values and determination of motor units number estimates (MUNE) in most mice. In contrast, in SOD1$^{G93A}$ mice that underwent OST, the significantly increased contractile response to ONS enabled clear separation of individual motor unit values (*Figure 5J*), enabling MUNE values to be determined (*Figure 5—figure supplement 5D*). Furthermore, as we previously reported in nerve-ligated WT mice, 22 delivery of repetitive ONS pulses (250ms bursts of 20 Hz illumination, every 1 s, for 180 s duration) to induce rhythmic, submaximal contraction of the TS muscle did not induce rapid muscle fatigue, whereas equivalent pulses of ENS stimulation of the contralateral TS muscle did result in rapid muscle fatigue (*Figure 5K*). This observation has significant implications for the ability to control repetitive motor functions in ALS patients.

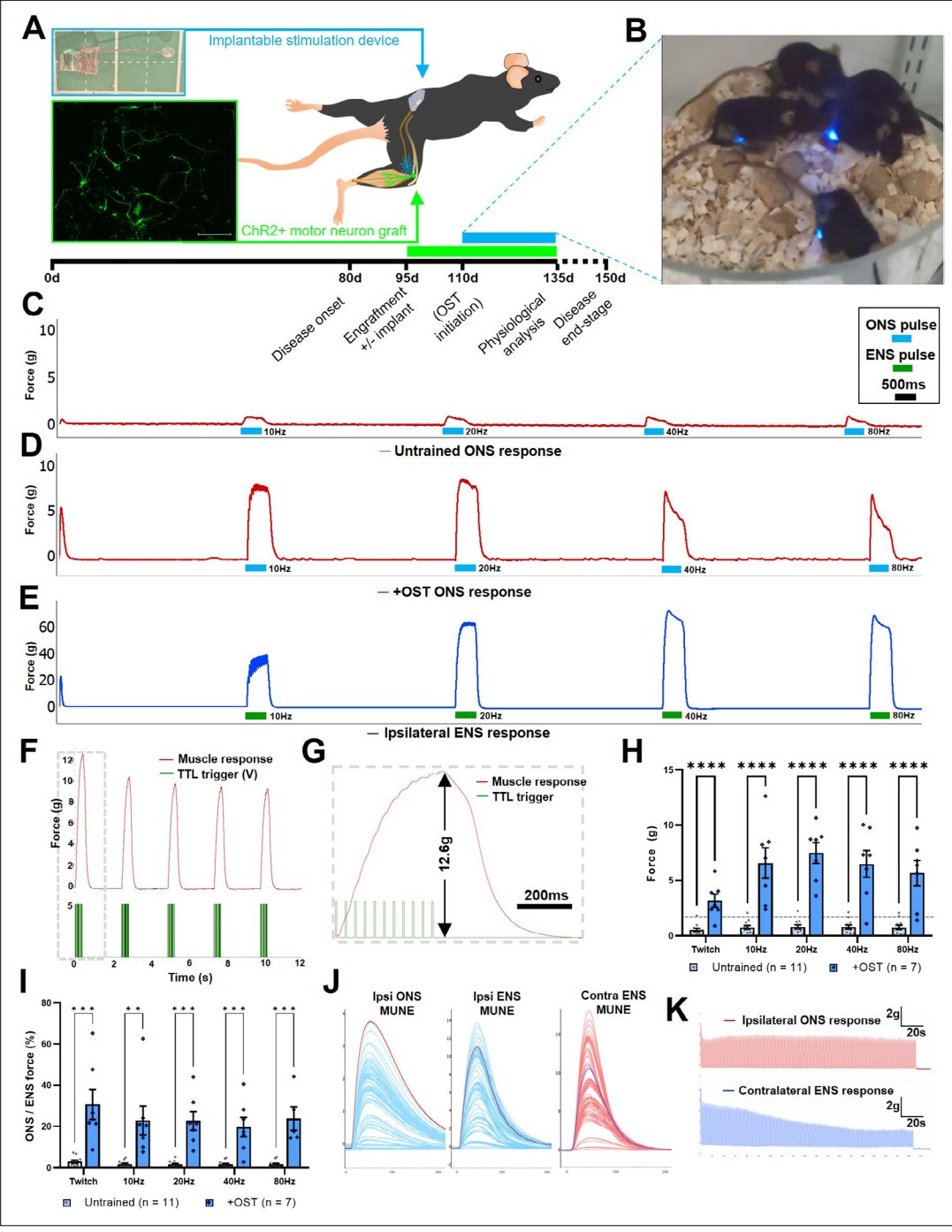

**Figure 5.** Daily optical stimulation training (OST) of post-symptom onset SOD1[G93A] mice engrafted with ChR2[+] motor neurons, significantly enhances contractile response to optical stimulation. (**A**) Schematic indicating intraneural engraftment site in distal tibial nerve and reinnervated triceps surae (TS) muscle, along with stimulation device (top inset) implantation site and subcutaneous LED position;+4 div MMC-treated ChR2[+] motor neurons express YFP (green; lower inset box); experimental timescale is shown below. (**B**) Still frame (taken from *Video 5*) showing daily OST. Representative isometric muscle tension physiology recordings from the TS muscle in response to specified pulses of ONS in untrained (**C**) and + OST (**D**) late-stage SOD1[G93A] mice, along with ENS response from the same muscle (**E**). (**F**) Delivery of an optimized pulse pattern elicits maximal response to ONS that can be used to finely control repetitive contractions; (**G**) Dashed box is shown at higher temporal resolution to indicate square-wave TTL pulse pattern (that drives LED stimulator) and an individual tetanic contraction. (**H**) Quantification of maximum contractile responses to indicated pulse patterns of acute ONS shows a highly

*Figure 5 continued*

significant improvement in force generation in late-stage SOD1[G93A] mice that underwent OST versus untrained controls (dashed horizontal line indicates maximum value from our previous study in nerve ligated WT mice). (**I**) The proportion of total muscle capacity (determined by supramaximal ENS minus ONS value) elicited by acute ONS is also significantly higher following OST; data represent mean ± SEM. (**J**) Motor unit number estimate (MUNE) traces obtained from a representative late-stage SOD1[G93A] mouse, following OST, in response to ipsilateral ONS and ENS, along with contralateral ENS (note, different scales). (**K**) Fatigue trace recordings comparing ipsilateral ONS (top) and contralateral ENS (bottom) in the same late-stage SOD1[G93A] mouse, in response to 250ms 20 Hz pulses, repeated every 1 s for 180 s.

The online version of this article includes the following figure supplement(s) for figure 5:

**Figure supplement 1.** An existing implantable device underwent minor modifications to improve suitability for optical stimulation training experiments.

**Figure supplement 2.** Light power recording and stimulation pattern recordings used to elicit acute optical nerve stimulation (ONS) throughout study.

**Figure supplement 3.** Optimization of optical nerve stimulation (ONS) pulse width to evoke maximum twitch contractile force.

**Figure supplement 4.** Optimization of optical nerve stimulation (ONS) pulse pattern to evoke maximum tetanic contractile force.

**Figure supplement 5.** Daily optical stimulation training (OST) in SOD1[G93A] mice does not affect muscle contractile characteristics in response to acute optical nerve stimulation (ONS).

**Figure supplement 6.** Comparison of optical nerve stimulation (ONS) versus electrical nerve stimulation (ENS) in late-stage SOD1G93A mice shows that supramaximal ENS still induces stronger contractile force, even after optical stimulation training (OST).

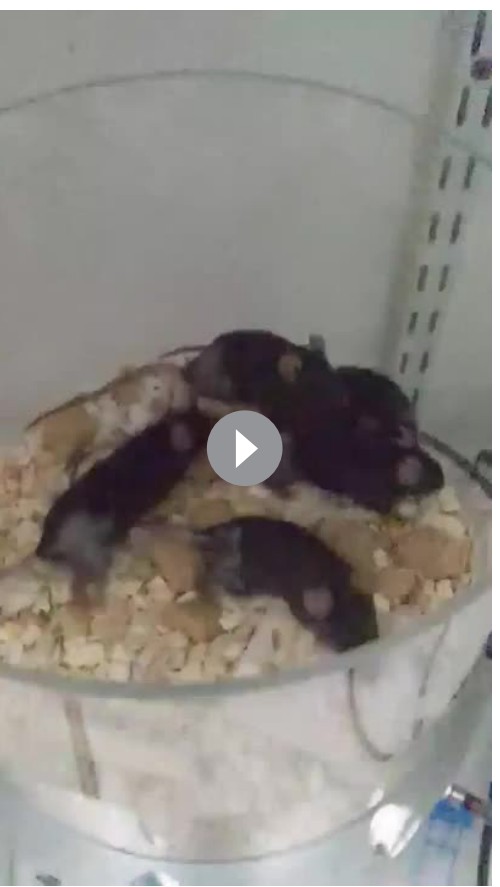

**Video 5.** Daily optical stimulation training significantly enhances elicited muscle force in SOD1[G93A] mice.
https://elifesciences.org/articles/88250/figures#video5

## Optical stimulation training prevents atrophy of reinnervated muscle fibres

Finally, having established that optical stimulation training significantly enhances the maximal force elicited by optical stimulation of engrafted ChR2[+] motor neurons in late-stage SOD1[G93A] mice, we examined whether long-term optical stimulation training could also prevent atrophy of reinnervated muscle fibres. Since NMJs comprise an extremely small volume of the entire muscle, high-resolution 3D imaging of the entire muscle to determine muscle fibre innervation status and fibre diameter information is not feasible. Therefore, we developed a novel technique, termed digital Cross-sectional area Analysis from Longitudinal Muscle Sections (dCALMS), in order to assess these properties. This involved 3D reconstruction and analysis of regions of interest, obtained from 30-µm-thick longitudinal TS muscle sections (*Figure 6A*) from ChR2[+] motor neuron engrafted, late-stage SOD1[G93A] mice that had undergone OST. Each region contained at least one NMJ innervated by a ChR2[+] motor neuron, along with randomly captured neighbouring fibres (*Figure 6B*, *Figure 6—figure supplement 1* and *Video 6*). The 3D reconstructions were then digitally re-sliced in the transverse orientation (*Figure 6C*, top panel), in order to obtain data on muscle fibre cross-sectional area (CSA) using

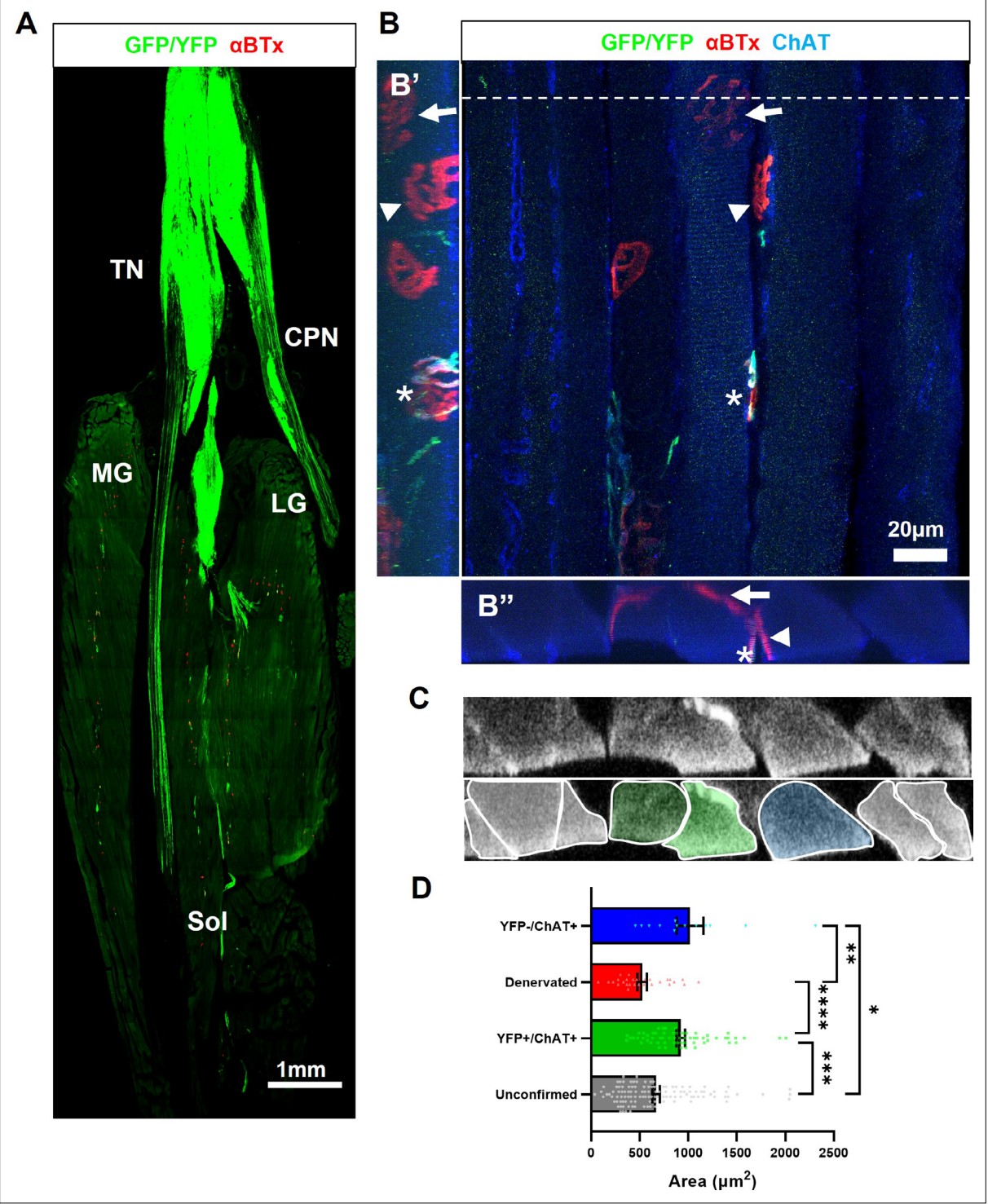

**Figure 6.** Optical stimulation training prevents atrophy of muscle fibres that have been reinnervated by ChR2[+] motor neurons in late-stage SOD1[G93A] mice. (**A**) Confocal tile-scan showing a single longitudinal section through the triceps surae muscle of 135d SOD1[G93A] mouse (35d post-engraftment), following daily OST; endplates (labeled with αBTx) innervated by GFP/YFP[+] engrafted motor neurons are evident throughout the whole muscle group, including fast-twitch medial gastrocnemius (MG) and lateral gastrocnemius (LG) muscles and the slow-twitch soleus (Sol) muscle. (**B**) Representative top-down maximum intensity projection (MIP) view of a confocal z-stack through a 30μ m longitudinal section obtained from the same mouse; including side-on (**B'**) and end-to-end (**B"**) MIP views of the same z-stack; a de novo NMJ, innervated by a ChR2[+] motor neuron (asterisk), is indicated on a muscle fibre that also has a denervated endplate (arrow), along with another muscle fibre still innervated by an endogenous choline-acetyltransferase (ChAT[+]) positive, GFP-negative motor neuron (arrowhead). (**C**) A digital slice (top) through the 3D z-stack obtained at the y-axis plane indicated by dashed line in

*Figure 6 continued on next page*

*Figure 6 continued*

(**B**) and colorized masks delineating individual muscle fibres (bottom); see *Video 6*. (**D**) Average cross-sectional area of individual muscle fibres with an innervation status that was unconfirmed (117 fibres), innervated by GFP +motor neurons (62 fibres), denervated (28 fibres), or innervated by endogenous motor axons (13 fibres); n=3 late-stage engrafted SOD1$^{G93A}$ mice that had undergone OST; Data shown as mean; error bars = SEM; one-way ANOVA with Tukey's post hoc correction: *denotes p ≤0.05; ** denotes p ≤0.0002; *** denotes p ≤0.002; **** denotes p ≤0.00002.

The online version of this article includes the following figure supplement(s) for figure 6:

**Figure supplement 1.** Daily optical stimulation training (OST) appears to enhance the extent of innervation of end-plates by engrafted ChR2+ motor neurons.

---

a semi-automated process (*Figure 6C*, lower panel). The dCALMS analysis revealed that the average CSA of muscle fibres innervated by engrafted ChR2$^+$ motor neurons were similar in size to fibres still innervated by residual endogenous motor neurons (922.3 vs 1018.3 µm$^2$; p=≤0.85; *Figure 6D*). Importantly, the CSA of muscle fibres innervated by ChR2$^+$ motor neurons was significantly greater than fibres with completely denervated endplates (average CSA = 525.4 µm2; p=≤0.00001) or fibres whose innervation status could not be determined (average CSA = 668.8 µm2; p ≤0.0003), since the endplate was outside the scanned region of interest.

## Discussion

This study shows for the first time that replacement stem cell-derived motor neurons can robustly and reliably reinnervate target muscles in the highly aggressive SOD1$^{G93A}$ mouse model of ALS, even when engrafted after onset of overt symptoms; moreover, the restored innervation can be maintained even until extremely late-stage disease. Furthermore, our findings suggest that engrafted ChR2$^+$ motor neurons can not only provide an interface to safely and selectively control the function of targeted muscle but, also, that regular optical stimulation training (OST) can be used to: (i) reinforce connectivity between engrafted motor neurons and muscle fibres; (ii) significantly enhance the maximal force elicited by optical stimulation of the targeted muscle; and (iii) prevent atrophy of muscle fibres that are reinnervated by engrafted motor neurons. The highly significant improvements in muscle innervation, atrophy prevention and maximum contractile force, as a result of the daily OST regimen, confirms that stimulation-induced activity is necessary to maximize connectivity between engrafted motor neurons and their target muscles.

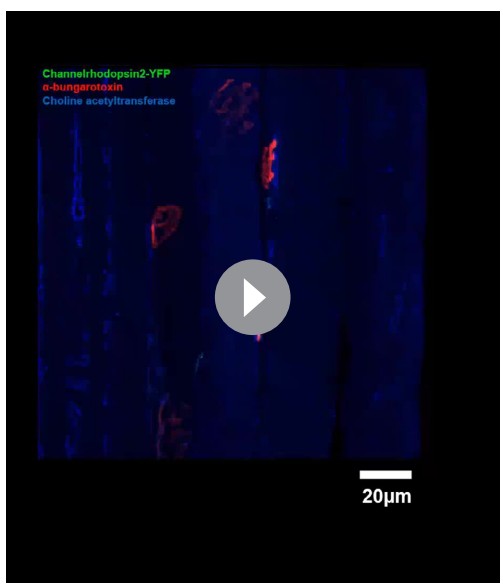

**Video 6.** 3D visualization of longitudinal muscle section from an engrafted SOD1$^{G93A}$ mouse along with 'dCALMS' muscle fiber analysis technique.
https://elifesciences.org/articles/88250/figures#video6

The prevailing view in the ALS field is that spinal motor neuron pathology first manifests at the nerve terminal and that muscle fibres are likely to actively contribute to degeneration of endogenous motor neurons in ALS (*Scaricamazza et al., 2021*). Our findings clearly demonstrate that affected muscles, even in the highly aggressive SOD1$^{G93A}$ ALS model, remain receptive to reinnervation by healthy engrafted motor neurons, even until late-stage disease. Moreover, once target muscles have been reinnervated, this approach enables implementation of a safe muscle training/ exercise regimen that can be used to preserve muscle integrity and prevent irreversible muscle wasting that otherwise occurs as a result of progressive neurodegeneration in ALS (*Mora, 1989*). Since skeletal muscles are not simply biomechanical actuators, but have complex functions in overall metabolic homeostasis, thermoregulation, venous return and maintenance of blood volume, the ability to prevent muscle atrophy using OST, is likely, by itself, to have major health benefits for ALS patients. In contrast, the use of electrical nerve stimulation (ENS) to

control muscle function or, indeed, ENS-based exercise programs is likely to accelerate degeneration of remaining motor axon terminals (*Guimarães-Costa et al., 2019*), and is therefore unlikely to be safe. Unlike ENS, the highly selective nature of ONS does not activate or interfere with endogenous motor neuron function and due to its ability to recruit motor units in physiological order, ONS has the added significant benefit of avoiding rapid muscle fatigue (*Llewellyn et al., 2010*). The ability to combat muscle atrophy, using OST, could extend the ability of targeted to execute functionally useful movements, potentially indefinitely in ALS patients.

Since differentiation methods that yield either fast-firing or slow-firing motor neurons did not appear to affect the ability of engrafted motor neurons to innervate the mixed fibre type TS muscle group in SOD1$^{G93A}$ mice, motor neuron subtype identity appears to be redundant in this case. Therefore, a single type of motor neuron, produced at scale, could potentially be used to target a large number of different muscles in each patient. This has advantages in terms of simplifying the regulatory approval process, since the donor motor neurons would be produced in exactly the same way, irrespective of the graft site or recipient.

Indeed, another key finding of this study that could streamline future translation of this therapeutic strategy, is the identification of a TCR-β targeting antibody, H57-597 mAb, as an effective mediator of allograft survival. Existing T-cell targeting monoclonal antibodies, such as OKT-3, have been clinically approved (*Page et al., 2013*) and, importantly, this form of immunosuppression overcomes severe adverse effects that we observed with the commonly used CNI-based immunosuppressant, tacrolimus (FK506). Our data shows that H57-597 mAb treatment was well tolerated during transient administration from symptom-onset up until late-stage disease in SOD1$^{G93A}$ mice, however, the aggressive disease progression in this model precludes investigation of longer-term effects and it remains to be seen how long-term TCR-β-based immunosuppression may be tolerated in ALS patients. A similar immunosuppression regimen involving a CD25 (IL2) targeting monoclonal antibody, Basiliximab, along with glucocorticoid and tacrolimus maintenance therapy, has been shown to be safe but poorly tolerated in ALS patients, as an independent investigative approach (*Fournier et al., 2018*), as well as part of a separate clinical trial assessing intraspinal grafts of neural precursor cells in ALS patients (*Mazzini et al., 2019*). Therefore, immunosuppression per se, is not an impediment to this strategy.

The use of allogeneic donor cells, with a safe and effective immunosuppression regimen, means that a future cell therapy could potentially be universally compatible with all ALS patients, which would significantly reduce costs and simplify the regulatory approval process, compared to individually tailored autologous cell grafts. Of course, further studies will be required to ensure that human-compatible, induced pluripotent stem cell (iPSC)-derived donor motor neurons are able to function in the same manner as allogeneic mouse ESC-derived motor neurons. The generation of HLA-matched super donor hiPSC lines may further mitigate the need for immunosuppression (*Turner et al., 2013*), however, immunogenicity of the ChR2 protein could mean that some form of immunosuppression may be necessary for any optogenetic therapy in the peripheral nervous system, including viral delivery (*Maimon et al., 2018*).

The anatomical separation of specific nerve branches in humans means that this approach could be used to target and independently control large numbers of muscles in each individual ALS patient. However, it will be first necessary to demonstrate safety and efficacy in a relatively low risk muscle to restore a simple motor function. For example, the common peroneal nerve is highly accessible from a neurosurgical perspective and existing ENS devices, developed to correct foot drop for other neurological disorders (*Hausmann et al., 2015*), could be readily adapted to assist ambulatory function in early-stage ALS patients. In the longer term, existing multichannel ENS devices, which have been developed to control more complex ADLs in high-level spinal cord injury (SCI) patients (*Memberg et al., 2014*), could also be adapted into a minimally invasive, transcutaneous optical stimulation device (*Maimon et al., 2017*). This combinatorial therapeutic strategy, comprising allogeneic donor cells, an effective immunosuppression regimen and optical stimulation device, is highly compatible with the rapidly evolving field of brain computer interface (BCI) technology. BCI could be used to decode a paralyzed patient's intention to perform a given movement in order to control the activity of the engrafted motor neurons, which provide the necessary interface to execute the intended movement via a wearable optical stimulation device. This novel approach would entirely bypass the severe damage that occurs throughout the entire CNS in ALS patients and enable autologous control of

movement. Moreover, this strategy also has broad utility for a wide range of other neurogenic causes of paralysis, such as spinal cord injury and stroke.

Although the findings of this study clearly demonstrate that our combinatorial cell therapy approach is effective in a highly aggressive mouse model of ALS, further investigation is required in order to confirm that the strategy can be applied to alternate model of ALS, with a longer lifespan, in order to fully explore the long-term efficacy of the approach, particularly in terms of chronic allograft survival using a transient immunosuppression approach. It is possible that some form of maintenance therapy may also be required to confer long term graft survival. Of course, the biggest challenge will be to demonstrate that human optogenetically-modified motor neurons, derived from either induced pluripotent stem cells (iPSCs) or human ESCs, are capable of reinnervating target muscles in the same manner as we have demonstrated for mESC-derived motor neurons. It will also be necessary to scale this cell therapy strategy up, using larger animal models that more accurately recapitulate human-scale anatomy.

Despite these remaining challenges, the findings of this study provide strong support for this novel cell therapy, which, if successful, could finally begin to deliver major health benefits for ALS patients.

## Materials and methods

Detailed methods are provided in the Supplementary methods.

### mESC motor neuron differentiation

The Channelrhodopsin2-YFP expressing mESCs (Clone C9G) used in this study were generated as previously described (*Bryson et al., 2014*). Motor neuron differentiation was performed according to a standard protocol developed by *Wichterle et al., 2002* for production of predominantly slow-firing medial motor column (MMC) identity motor neurons and an updated 'caudalized-ventralized' (CV) protocol, also developed by the Wichterle group (*Peljto et al., 2010*), that produces a higher proportion of motor neurons with fast-firing properties. Following differentiation on Day 5 (or Day 7 for the C-V protocol), embryoid bodies (EBs) containing differentiated motor neurons were dissociated and total cells were resuspended in PBS at a concentration of 50,000 cell/µl. Where indicated, mitomycin-C (1µg/µl) was added to the EBs for 2 hr prior to dissociation. Nile Blue A (0.0002% final concentration) was added to the cell suspension and the cells were kept on ice until engraftment. Single nucleotide polymorphism (SNP) analysis of mESC lines (Clone C9G, HBG3 and a C57BL/6 J mESC line for reference) was carried out by Charles River Laboratories. See the Supplementary methods for full details.

### Mice

All procedures and experiments involving animals were carried out under License from the UK Home Office in accordance with the Animals (Scientific Procedures) Act 1986 (Amended Regulations 2012), following ethical approval from the UCL Queen Square Institute of Neurology Animal Welfare Ethical Review Body (AWERB), and in accordance with the ARRIVE guidelines. B6.Cg-Tg(SOD1*G93A)1Gur/J mice (The Jackson Laboratory, stock number 004435) were bred specifically for this study by mating presymptomatic male transgene carriers with congenic C57BL/6 J females (Charles River Laboratories).

### Intraneural engraftment of mESC-derived ChR2+ motor neurons

Surgical engraftment of ChR2[+] motor neurons was performed under aseptic conditions, as previously described (*Bryson et al., 2014*). Briefly, 1µ l of dissociated EB cell suspension (50,000 cells) was injected into the tibial nerve close to the trifurcation point of the sciatic nerve, using a 5µ l Hamilton syringe equipped with a customized 33 G needle. Where indicated, an implantable optical stimulation device was inserted through the same surgical incision and positioned subcutaneously under the skin on the back; the trailing LED was fixed with sutures to the overlying muscles at the graft site during wound closure. Immunosuppression, as indicated, was initiated at the time of surgical engraftment. See the Supplementary methods for full details.

### Implantable optical stimulation devices and power transmission system

Optical stimulation devices were largely produced as described by *Montgomery et al., 2015*, with minor, but essential, modifications to the PCB design and encapsulation method. Similarly, the power transmission

system used to activate the implanted LED devices was also largely as described by, however, a Solid State Switch (MiniCircuits; ZX80-DR230-S+), controlled by a USB-TTL Interface (Prizmatix), was used to generate specific optical stimulation patterns. Engrafted SOD1$^{G93A}$ mice that underwent daily optical stimulation training were placed in the stimulation chamber for 1 hr/day from post-engraftment day 14 until termination of the experiment at late-stage disease (132.4±6.8 days). See the Supplementary methods for full details.

### Isometric muscle tension physiological analysis

At the experimental end-point, engrafted SOD1$^{G93A}$ mice underwent isometric muscle tension physiology, in order to accurately determine the contractile properties of the triceps surae muscle in response to acute optical stimulation, as previously described (*Bryson et al., 2014*), with the following modifications: a PowerLab 4/30 stimulation and recording system (AD Instruments) was used to deliver bespoke electrical stimulation signals, either as direct constant voltage pulses applied to the nerve via bipolar electrodes for electrical nerve stimulation (ENS), or used as a 5 V TTL signal to activate a 470 nm LED light-source (CoolLED; pe100), delivered to the exposed sciatic nerve via a liquid lightguide for optical nerve stimulation (ONS); stimulation program available on request. LabChart software (AD Instruments) was used for automated data analysis of contractile parameters. See the Supplementary methods for full details.

### Nerve and muscle histology and image analysis

See the Supplementary methods for full details, including automated motor/sensory axon CSA analysis, axon counts and innervation analysis method. The digital CSA analysis of longitudinal muscle sections (dCALMS) method is also described in full in the Supplementary methods.

### Quantification and statistical analysis

#### Sample sizes

The number of mice (n) is provided in the figures and/or figure legends; also see *Appendix 1—table 2* and *Appendix 1—table 3*.

#### Statistical analysis

All data are presented as mean ± SEM unless otherwise indicated. GraphPad Prism 9 (Prism) was used for statistical analyses. No out-liners or data points were eliminated. Differences between two groups were assessed using multiple two-tailed unpaired t tests. Differences between more than two groups were assessed by using one-way or two-way analysis of variance (ANOVA) with multiple comparison correction, or mixed model effects analysis, as stated in the figure legends. Significance was defined as *p ≤0.05, **p ≤0.01, ***p ≤0.001, or ****p ≤0.0001. See the Supplementary methods for further details.

## Acknowledgements

The authors are grateful to Profs Elizabeth Fisher, Jennifer Morgan (UCL) and Victor Tybulewicz (The Crick Institute) for helpful discussions about immuno-compatibility of allogeneic cells in mice and adverse effects of CNIs on muscle, and Dr Henry Lancashire for assistance with modified PCB production. In addition, we are grateful to Prof Ada Poon and Dr Yuji Tanabe (Stanford University) for assistance with establishing their wireless optical stimulation system. JBB was supported by a Motor Neurone Disease Association Lady Edith Wolfson Senior Non-clinical Fellowship (Bryson 965–799), as well as an NIHR BRC UCL Excellence Fellowship award (BRC371/ED/AT/101310), in addition to grant support from the Rosetrees Trust (ref: M643) and an Early Career Researcher Award from the Richard Stravitz Foundation. This study was supported in part by the UK Medical Research Council (MRC) research project grant (MR/R011648/1), awarded to JBB, AD and LG (principal investigator) and a Thierry Latran Foundation project grant (JBB and LG). LG is supported by Brain Research UK and holds The Graham Watts Senior Research Fellowship (LG).

## Additional information

### Funding

| Funder | Grant reference number | Author |
|--------|------------------------|--------|
| Motor Neurone Disease Association | Bryson 965-799 | J Barney Bryson |
| UCLH Biomedical Research Centre | BRC371/ED/AT/101310 | J Barney Bryson |
| Rosetrees Trust | M643 | J Barney Bryson |
| Richard Stravitz Foundation | | J Barney Bryson |
| Medical Research Council | MR/R011648/1 | J Barney Bryson<br>Andreas Demosthenous<br>Linda Greensmith |
| Thierry Latran Foundation | | J Barney Bryson<br>Linda Greensmith |
| Brain Research UK | | Linda Greensmith |

The funders had no role in study design, data collection and interpretation, or the decision to submit the work for publication.

### Author contributions

J Barney Bryson, Conceptualization, Data curation, Formal analysis, Supervision, Funding acquisition, Validation, Investigation, Visualization, Methodology, Writing – original draft, Project administration, Writing – review and editing; Alexandra Kourgiantaki, Formal analysis, Investigation, Methodology; Dai Jiang, Andreas Demosthenous, Resources, Methodology; Linda Greensmith, Conceptualization, Resources, Writing – review and editing

### Author ORCIDs

J Barney Bryson ⬤ https://orcid.org/0000-0001-5486-0258
Linda Greensmith ⬤ https://orcid.org/0000-0002-7839-5052

Reviewer #1 (Public Review): https://doi.org/10.7554/eLife.88250.3.sa1
Reviewer #2 (Public Review): https://doi.org/10.7554/eLife.88250.3.sa2
Author Response https://doi.org/10.7554/eLife.88250.3.sa3

# Additional files

### Supplementary files

• MDAR checklist

### Data availability

The raw data that supports the findings of this study is available through the UCL Research Data Repository (https://doi.org/10.5522/04/c.6953886).

The following dataset was generated:

| Author(s) | Year | Dataset title | Dataset URL | Database and Identifier |
|-----------|------|---------------|-------------|-------------------------|
| Kourgiantaki A, Jiang D, Demosthenous A, Greensmith L | 2023 | Data in support of 2023 publication in eLife: An optogenetic cell therapy to restore control of target muscles in an aggressive mouse model of Amyotrophic Lateral Sclerosis | https://doi.org/10.5522/04/c.6953886 | UCL Research Data Repository, 10.5522/04/c.6953886 |

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

## Appendix 1

### Supplementary methods

#### Cell culture

The mESCs used in this study were generated as previously described (*Bryson et al., 2014*). Briefly, mESC underwent successive rounds of gene-targeting to insert a Hb9::GFP-IRES-CD14 reporter construct to enable motor neuron purification, insertion of a CAG::Channelrhodopsin2-YFP (ChR2-YFP) construct using Tol2 transposition to enable optogenetic control of neural activity (*Nagel et al., 2003*; *Boyden et al., 2005*) and insertion of a CAG::Glial derived neurotrophic factor (Gdnf) construct using PiggyBac targeting to promote motor neuron survival; this resulted in generation of clone C9G. Undifferentiated mESCs, from clone C9G, were expanded on irradiated C57BL/6 mouse embryonic fibroblasts (iMEFs) (Gibco Cat # A34965, RRID:CVCL_RB07) according to standard cell culture protocols. For maintenance, briefly, freshly thawed mESC aliquots were plated on iMEF feeder layers, grown on gelatinized 24-well or 6-well plastic tissue culture treated dishes (approx. 2.5x105 cells/cm²), in mESC media, consisting of Knockout-DMEM (Gibco), supplemented with 15% ES-qualified FBS, 1 x nucleosides, 1 x non-essential amino acids, 1 x L-glutamine + penicillin/streptomycin, 0.1 mM 2-mercaptoethanol (all supplied by Gibco) and 100 U/ml ESGRO Recombinant Mouse LIF Protein (Merck Millipore); cells were incubated at 37 °C, 5% CO2, cells were fed daily and passaged or differentiated prior to colonies reaching 70% confluence.

#### Motor neuron differentiation

Motor neuron differentiation was carried out as previously described (*Wichterle et al., 2002*; *Peljto et al., 2010*). Briefly, mESC colonies were dissociated using TryplE solution (Gibco), 7 mins at 37 °C, and transferred to non-tissue culture treated 10 cm plates in differentiation medium, consisting of a 1:1 mix of advanced-DMEM/F12 and neurobasal media, 10% knockout serum replacement (KO-SR), 1 x L-glutamine/Penicillin/Streptomycin and 0.1 mM 2-mercaptoethanol (hereafter, termed ADNFK media; all components source from Gibco). For generation of medial motor column (MMC) identity motor neurons, mESCs (20,000 cell/ml) were incubated for 2 days at 37 °C, 5% CO2 as floating embryoid bodies (EBs) in ADFNK, before being split 1:2 in ADFNK media supplemented with 1µ M retinoic acid (RA; Sigma) and 0.5µ M smoothened agonist (Sag; Sigma) between day 3–5 of differentiation. For generation of lateral motor column (LMC) identity motor neurons the modified 'caudalized-ventralized' (C-V) protocol was used (*Peljto et al., 2010*) briefly, mESCs (10,000 cell/ml) were incubated for 2 days at 37 °C, 5% CO2 as floating embryoid bodies (EBs) in ADFNK, before being split 1:2 in unsupplemented ADFNK media between days 3 and 5 of differentiation and then transferred to ADFNK supplemented with 1µ M RA between differentiation days 5 and 7. On differentiation day 5 or 7, differentiated EBs were pre-treated with 1µ g/ml of mitomycin-C for 2 hr (where indicated), washed in L15 media and dissociated in Accumax (Thermo Fisher Scientific), washed in L15 media, passed through a 20µ m cell strainer and collected in PBS at a final concentration of 50,000 cells/µl; differentiated motor neurons were kept on ice prior to in vivo engraftment.

#### Single nucleotide polymorphism (SNP) analysis of mESC lines

mESCs derived from clone #C9G, along with a C57BL/6 J mESC line and the commonly used HBG3 (Hb9::GFP) mESC line, as controls, were grown to confluence in the absence of feeder cells, in mESC media and cells were collected and frozen as dry cell pellets, before being sent for commercial SNP analysis (Charles River Laboratories).

#### Production of implantable optical stimulation devices

Optical stimulation devices were largely produced as described by *Montgomery et al., 2015*, with the following modifications: the design of individual printed circuit boards (PCBs) was modified to enlarge the central pads and the PCB was then fabricated on 0.2-mm-thick FR4 board, which greatly facilitated assembly and mounting of individual components (*Figure 5—figure supplement 1*) and Supplementary CAD files; an alternative 470 nm (145mcd) LED (Wurth Elektronik, Manufacturer Part No: 150120BS75000) was attached to 1.5 cm paired trailing wire connected to the power-receiving rectifier circuit, again to facilitate assembly; a 3-turn power-receiving coil with an internal diameter of 1.7 mm was used. Following positioning of Schottky diodes and capacitors on the PCB, and after coating with lead-free solder paste (Chip Quick), brief exposure to a flame from a gas soldering iron was used in lieu of reflow soldering. Fully assembled devices were cleaned in an ultrasonic water bath, rinsed in 100% ethanol, dried and then encapsulated by insertion into a silicone-grease

coated 0.2 ml PCR tube that was then partially filled with optically clear epoxy (Opti-tec 5001) to cover the device. The epoxy was cured for 1 hr at 65 °C and remaining uncured epoxy was then applied to the inverted trailing LED to form a droplet, before additional curing at 65 °C. After encapsulation, devices were removed from 0.2 ml PCR tubes, cleaned again in an ultrasonic water bath (to remove residual silicone grease), before being immersed in 100% ethanol for sterilization. Modification of the encapsulation method was necessary to ensure longevity of the device following in vivo implantation. Prior to implantation, devices were transferred to sterile PBS and tested in the RF-resonance chamber (see below) to ensure they were fully functional.

## Radio Frequency (RF) resonance cavity power transmission system

As above, the power transmission system used to activate the implanted LED devices was largely as described by *Montgomery et al., 2015*, with the following modifications: a SynthNV RF Signal Generator (WindFreak) was used to generate a constant 1497MHz RF output (1-2bBm), that was fed into an absorptive SPDT, Solid State Switch (MiniCircuits; ZX80-DR230-S+), powered by a 1.5 V CR2032 coin battery, which was controlled by a USB-TTL Interface (Prizmatix), to generate specific optical stimulation patterns (*Figure 5—figure supplements 2–4*).

## Animals

B6.Cg-Tg(SOD1*G93A)1Gur/J mice (JAX RRID:MGI:2181028) were bred specifically for this study by mating presymptomatic male transgene carriers with congenic C57BL/6 J females (Charles River Laboratories); progeny were genotyped by standard PCR analysis and heterozygous transgenic mice were housed under a 12 hr dark/light cycle in IVC cages with behavioral enrichment and ad libitum access to food and water, until they reached the appropriate age. A total of 73 SOD1$^{G93A}$ mice and 17 non-transgenic wild-type littermates were directly used in this study; an additional 129 SOD1$^{G93A}$ mice were used to develop the methodology.

All procedures and experiments involving animals were carried out under License from the UK Home Office in accordance with the Animals (Scientific Procedures) Act 1986 (Amend-ed Regulations 2012) and following ethical approval from the UCL Queen Square Institute of Neurology Animal Welfare Ethical Review Body (AWERB). We used the ARRIVE checklist when writing our report *Percie du Sert et al., 2020*; see below:

## Study design parameters
### Group size calculation
As this was a success/failure study, sample sizes were not predetermined for in vivo engraftment studies, rather we aimed for a repeated success rate (ie. target muscle innervation by engrafted motor neurons). However, in order to enable us to undertake robust statistical analysis of our findings, we therefore aimed minimum group size of 6–8, and we performed retrospective power calculations to confirm that this was an appropriate sample size.

### Inclusion/exclusion criteria
No data were excluded from the analyses. In rare cases, failure of the implantable LED device (used to deliver optical stimulation training; OST) before the start of the training period resulted in some mice being reassigned from the OST group to the 'untrained' group; a detailed record of visual confirmation of LED device functionality was maintained for all stimulated mice to ensure the device was fully functional throughout the training period.

### Replication
In vivo engraftment studies, physiological analysis and histological analysis were replicated in n=32 SOD1$^{G93A}$ mice for the major findings of this study, over a 2-year period, using multiple batches of ESC-derived motor neurons that were specifically prepared for each cohort (typically 4–8 mice per cohort); experimental outcomes were highly consistent and reproducible. Antibody-based immunosuppression conferred 100% graft survival in over 84 SOD1$^{G93A}$ mice to date, with positive responses to optical stimulation (determined by physiological analysis).

### Randomization
At the start of the study, animals in each litter (of appropriate genotype) were randomly assigned to each experimental cohort. All mice received grafts of the same ChR2$^+$ motor neurons that were

differentiated to coincide with each cohort of mice reaching the appropriate age for enrollment in the study. Rather SOD1$^{G93A}$ mice were custom bred for each phase of the project, which naturally evolved as specific experimental obstacles arose and were overcome (e.g. toxicity of first choice immunosuppressant); these mice were assigned to specific groups following genotype confirmation (to reduce the number of experimental mice, since wild-type littermates were not the primary focus of this study) and upon reaching the appropriate age; isometric muscle tension physiological analysis in response to optical (and electrical) stimulation was used as the primary outcome measure, in order to determine the absolute maximum stimulus-evoked muscle force; this outcome measure was selected for its accuracy, reproducibility and reliability based on our past experience in pre-clinical therapeutic development.

## Blinding

In vivo engraftment and physiological analysis were blinded where possible. Since all mice were the same genotype and received the same cell grafts the experimenter was therefore aware of this, however, the experimenter was blinded to stimulation conditions (ie. untrained versus OST groups) for physiological analysis. Importantly, physiological recordings of maximal muscle force provides a highly reliable functional readout, moreover, automated software was used to extract muscle contractile characteristics data to eliminate bias. Histological analysis was performed blind for nerve and muscle analyses, with the exception of dCALMS analysis, since only SOD1$^{G93A}$ mice that had undergone optical stimulation training were used for this analysis. As reported in the methods, automated or semi-automated analysis was used for all histological analysis to eliminate bias.

## In vivo intraneural engraftment of mESC-derived ChR2 + motor neurons

Mice received prophylactic analgesia (buprenorphine, 0.1 mg/kg), during induction of anaesthesia and preparation of the surgical site. The immunosuppressant monoclonal antibody, H57-597 mAb (Bio X Cell Cat# BE0102, RRID:AB_10950158) was also administered at the start of the surgical procedure (day 0) by intraperitoneal (i.p.) injection (1 mg/kg, diluted in 100µ l of sterile saline), which was repeated on days 1, 3, 7 and 14 post-engraftment. Alternatively, in a subset of mice (n=15), immunosuppression was achieved by daily i.p. administration of FK506 tacrolimus; 5 mg/kg/day, note: due to insolubility in aqueous solution, FK506 (16.67 mg/ml) was dissolved in DMSO and stored as frozen 100µ l aliquots that were freshly thawed and prepared immediately prior to injection by dilution in 30µ l of 100% ethanol (15% final concentration) and 70µ l sterile PBS containing 2% Tween-20 (8.33 mg/ml final concentration of FK506; 0.6µ l/g of body mass, mixed with saline to a final volume of 125µ l immediately prior to i.p. injection). Mice were closely monitored and weighed at the time of immunosuppressant administration. As previously reported, H57-597 induced acute loss of body mass, which fully recovered in all cases within 7 days from the initial dose; this adverse effect can be blocked by pre-treatment with cyclosporin A (*Murakami et al., 1995*).

Intraneural engraftment of ChR2$^+$ motor neurons was performed under deep anaesthesia, using aseptic surgical techniques and body temperature was maintained throughout the surgical procedure using a 37 °C heat mat, as previously described (*Bryson et al., 2014*). Briefly, under a stereoscopic operating microscope (Zeiss), a 3 mm skin incision was made in the lower posterior thigh and superficial muscles were separated and retracted to expose the sciatic nerve, immediately proximal to the trifurcation point in the popliteal fossa. The sciatic nerve was elevated from underlying muscles using a curved non-cutting suture needle, placing it under slight tension, and a 27 G needle was used to make a small incision in the epineurium of the tibial and common peroneal nerve branches. Immediately prior to intraneural engraftment ChR2$^+$ motor neurons (50,000 cells/µl in PBS containing 0.0004% Nile Blue dye, maintained on ice) were carefully resuspended by pipetting and loaded into a 5µ l Hamilton syringe, equipped with a customized 33 G needle. The needle was then carefully inserted through the incision made in the epineurium and guided along the length of the nerve (approximately 3–4 mm), to the trifurcation point of the sciatic nerve, in order to prevent back-pressure induced leakage of cells; 1µ l of cell suspension was slowly injected into the tibial nerve and 0.5µ l into the common peroneal nerve. Visualization of Nile Blue labelled cells was used to verify intraneural cell delivery. In procedures during later stages of this study, a fine-tipped (approximately 100µ m outer diameter) pulled glass microinjection needle, with a beveled tip and customized positive displacement plunger (Alpha laboratories) was used for intraneural injection of cells, to minimize damage to the nerve. After injection, the needle was carefully withdrawn and the sciatic nerve was returned to its normal position. A subset of mice (n=7; excluding animals in which the device failed during the stimulation period) underwent implantation of an optical stimulation device

(described above). Briefly, fine round-tipped tissue forceps were inserted through the same skin incision, along the length of the thigh bone and rostrally along the back, to create a subcutaneous cavity. The main body of the sterilized device was inserted into the cavity and positioned close to, and parallel with, the lower thoracic spine (to assist alignment of the power receiving coils with the RF-resonance frequency power transmitter). The trailing wire and encapsulated LED was secured in place immediately superficial to the graft site in the sciatic nerve by 8–0 non-absorbable sutures that were also used to return the overlying muscles to their normal position. Finally, the skin incision was closed using 8–0 absorbable sutures and post-operative mice were transferred to a heated chamber and allowed to fully recover before being returned to their home-cage. Mice were closely observed for 10 days following surgery.

## Daily optical stimulation training

After 14d post-engraftment (to allow sufficient time for engrafted motor neurons to extend axons far enough to reach target muscles), mice with implanted stimulation devices were transferred to a round acrylic chamber, containing clean bedding and nesting material, positioned on top of the RF-resonance cavity (*Figure 5A and B* and *Figure 5—figure supplement 1C*). Where possible, multiply-housed littermates were transferred to the chamber at the same time. A red acrylic dome was placed in the centre of the chamber to encourage mice to spend time in that location, since resonance power transmission is less efficient at the periphery of the chamber. Engrafted SOD1$^{G93A}$ mice underwent daily optical stimulation (pulse pattern described below; *Figure 5—figure supplements 2–4*) for 1 hr/day from post-engraftment day 14 until termination of the experiment at late-stage disease (132.4±6.8 days).

## Isometric muscle tension physiological analysis

At the experimental end-point, engrafted SOD1$^{G93A}$ mice underwent isometric muscle tension physiology, in order to accurately determine the contractile properties of target muscles in response to acute optical stimulation, largely as previously described (*Bryson et al., 2014*), with the following modifications: a PowerLab 4/30 stimulation and recording system (AD Instruments) was used to deliver bespoke electrical stimulation signals, either as direct constant voltage pulses applied to the nerve via bipolar electrodes for electrical nerve stimulation, or used as a 5 V TTL signal to activate a 470 nm LED light-source (CoolLED; pe100), delivered to the exposed sciatic nerve via a liquid lightguide for optical stimulation; stimulation program available on request. As previously reported, percentage light power intensity can be controlled over a range of 0–80 mW/mm2; 50% (40 mW/mm2) supramaximal stimulation was used for most optically-evoked muscle force recordings, however, maximum motor unit activation was achieved using light intensities ≤2.5 mW/mm2. For electrical stimulation, 0.2ms supramaximal constant-voltage (5 V) pulses were directly applied to the exposed sciatic nerve via a bipolar stimulating electrode, as individual pulses (for twitch contractions) or at 20, 40, 80, and 100 Hz burst (0.5 s duration) with a 5 second rest interval to interrogate the full range of maximal tetanic contractile force values.

Distal tendons of individual target muscles were attached to 25 g or 55 g UF-1 force transducers coupled to a bridge amp (AD Instruments). LabChart software (AD Instruments) was used for automated data analysis of parameters including maximum twitch and tetanic contractile force, time-to-rise, time-to-peak and ½ relaxation time (during twitch contractions). Motor unit number estimation (MUNE) was performed using a bespoke pattern of rising and falling amplitude electrical stimuli (0–5 V, 0.2ms pulse width), repeated every 1 s (LabChart programme available upon request), or by 1 s interval 5 V TTL trigger pulses in combination with manual cycling of the percentage power output from the pe-100 (470 nm) light source (CoolLED) between 1–10%.

## Nerve and muscle histology

Immediately after physiological analysis, animals were euthanized and the ipsilateral (engrafted) sciatic nerve (SN), attached to the triceps surae (TS) muscle group, along with the contralateral SN and tibialis anterior (TA) and extensor digitorum longus (EDL) muscles, were dissected, rinsed in TBS and post-fixed in 4% paraformaldehyde in TBS for a minimum of 2 hr, followed by cryoprotection in TBS containing 20% sucrose for >12 hrs. Tissue was mounted in aluminium-foil moulds containing OCT (Tissue-Tek) and rapidly frozen on dry-ice, then stored at –20 °C prior to cryosectioning. Serial longitudinal sections from the ipsilateral SN and TS muscle, and TA/EDL muscles were cut at 30μ m thickness, encompassing the entire tissue block to enable 3D reconstruction of the extent

of reinnervation throughout the whole muscle (only performed in selected cases due to time- and resource-intensive nature of imaging data acquisition and processing). Transverse 10-µm-thick sections were cut from the contralateral SN for axonal analysis. For immunostaining, muscle/nerve sections were washed 3x5 min with TBS, blocked for 1 hr in TBS containing 0.2% Triton X-100 and 5% donkey serum and sections were double-labeled with the following primary antibodies, raised in either rabbit or goat, applied in combination overnight in TBS containing 0.2% Triton X-100 and 2% donkey serum: goat anti-choline acetyltransferase, 1:100 (Millipore Cat# AB144P, RRID:AB_2079751); goat anti-green fluorescent protein, 1:500 (Abcam Cat# ab6673, RRID:AB_305643); rabbit anti-green fluorescent protein, 1:500 (Molecular Probes Cat# A-11122, RRID:AB_221569); rabbit anti-choline acetyltransferase, 1:200 (Abcam Cat# ab178850, RRID:AB_2721842); rabbit anti-βIII tubulin; 1:500 (Covance Cat# PRB-435P-100, RRID:AB_291637). Secondary antibodies, raised in donkey were diluted 1:500, along with α-bungarotoxin-Alexa568/647 (1:500; Invitrogen) and DAPI (1:1000; Sigma) in TBS containing 0.2 Triton X-100 and 2% donkey serum and applied for 2 hr, before washing 3x5 min with TBS and coverslips were then mounted using fluorescent mounting medium (Invitrogen).

## Image processing

Image acquisition was performed using a Zeiss LSM780 confocal microscope, to acquire tile-scan images of entire SN/TS sections using a 20 x objective (pinhole set to 30µ m to obtain fluorescent signal from entire thickness of the section); high-resolution z-stacks from specific regions of interest were acquired using either 40 x or 63 x oil-immersion objectives. Images reported here were either prepared directly, using Zeiss Zen Blue/Black image processing software or processed and analysed using Fiji (ImageJ) and MetaMorph (Molecular Devices) software for 3D reconstruction, as follows: serial tile-scan images encompassing the whole SN/TS tissue block were converted into compressed tiff-format files, assembled into an image stack and individual planes were aligned using the StackReg plugin (*Thévenaz et al., 1998*), aligned stacks were then converted into rendered 3D reconstructions using MetaMorph software and saved as compressed AVI video files for visualization purposes (note: automated NMJ analysis, described below, was performed on individual image planes from each stack). Tile-scans were also acquired from contralateral SN sections, using a 40 x or 63 x oil-immersion objective, for automated axon cross-sectional area analysis.

## Automated endplate and innervation analysis

Serial tile-scan images, in RGB Color format, from the whole TS muscle from SOD1$^{G93A}$ mice at the experimental end-point, 135 days, were analysed as follows, using Fiji (ImageJ) software: endplates, labelled with α-bungartoxin conjugated to Alexa-568 or Alexa-647 were assigned to the red color channel and color thresholding, using consistent parameters (Hue = 0–17 [pass], Saturation = 0–255 [pass], Brightness = 75–255 [pass]) was performed to specifically identify endplates; the number of endplates in each plane was then quantified using the 'Analyze Particles' feature of ImageJ (key parameters include: Area = 50–500µ m, Circularity 0–1, Show "Overlay masks"). Quantification of innervated endplates was performed in a similar manner, using colour thresholding parameters to identify double-labelled endplates that stained positive for both α-bungarotoxin (assigned to red channel) and GFP/YFP + motor axons (assigned to the green channel), using the following parameters: Hue = 23–44; Saturation = 0–255; Brightness = 90–255. This automated analysis method is intended to quantify the total number of endplates and provide an indication of their innervation status; determination of full versus partial innervation would require much higher resolution imaging which is not feasible for the whole TS muscle. Additionally, severe muscle atrophy at late-stage disease decreases the TS muscle volume, thus the number and size of tile-scan images would be approximately threefold greater for wild-type mice.

## Motor and sensory axon nerve analysis

Individual channels, representing total axons (stained for β-III tubulin) and motor axons (stained for choline-acetyltransferase), from tile-scans of contralateral SNs were used to determine number and cross-sectional area (CSA) of each axonal type, to determine the effect of different immunosuppression regimens. Briefly, images were thresholded in ImageJ, using identical settings to delineate individual axons, individual branches were then manually circled, using the freehand selection tool, and axonal counts and CSA were automatically quantified using the 'Analyze Particles' tool.

## Digital CSA analysis of longitudinal muscle sections (dCALMS)

Briefly, z-stacks from regions of interest incorporating at least 1 GFP/YFP + motor axon terminal were acquired from the whole 30µ m thickness of individual TS muscle sections; the 'reslice' feature of ImageJ was then used to obtain digital transverse orientation images from each ROI to interrogate muscle fibre CSA throughout the y-plane of the image stack and correlation with the innervation status; triple labeling with GFP/YFP, α-bungarotoxin and choline-acetyltransferase labeling enabled the innervation status of adjacent fibres within the ROI to be assigned to the following categories: denervated (α-bungarotoxin only), innervated by endogenous motor neurons (α-bungarotoxin and choline-acetyltransferase), innervated by engrafted ChR2$^+$ motor neurons (α-bungarotoxin, choline-acetyltransferase and GFP/YFP) or unknown when no endplate was visible within the ROI. Overexposure of the channel used to acquire choline-acetyltransferase staining enabled visualization of the muscle CSA, which underwent gaussian blurring to aid definition of the sarcolemma (as shown in *Figure 6C*, colorized images versus non-blurred images shown in black and white). The edge of each muscle fibre at the level of the endplate (where present) in the y-plane (or the maximum diameter, where no endplate was evident), was manually circled using the freehand selection tool and analysed using the 'measure' function of ImageJ. ROIs were analysed from n=3 mice that underwent ChR2$^+$ motor neuron engraftment and optical stimulation training. Only fibres that were entirely within the z-plane were measured at their widest point along the y-plane (*Figure 6C* and *Video 6*); partial fibres, were excluded.

## Statistical analysis

All data are presented as mean ± SEM unless otherwise indicated. GraphPad Prism 9 (Prism) was used for statistical analyses. No out-liners or data points were eliminated. Differences between two groups were assessed using multiple two-tailed unpaired t tests. Differences between more than two groups were assessed by using one-way or two-way analysis of variance (ANOVA). Correction for multiple testing was performed as described in the figure legends. Significance was defined as *p ≤ 0.05, **p ≤ 0.01, ***p ≤ 0.001, or ****p ≤ 0.0001.

**Appendix 1—table 1.** Single Nucleotide Polymorphism (SNP) analysis of Clone #C9G and control mESC lines to confirm genetic background strain.

SNP analysis confirmed that mESC Clone #C9G, used in this study, originated from a different genetic background compared to host mice (C57Bl/6J background strain). C57BL/6J mESCs and HBG3 mESCs were included as controls.

**Conformity of Sample to Reference Strain Allelic Profile**

| Sample ID-Code | Reference | # Called | Call Rate | Percent Match | Percent Het |
|---|---|---|---|---|---|
| 001-Clone C9G mESCs | 129S1SvlmJ | 382 | 99.5% | 97.5% | 0.3% |
| 002-C57BL/6 J mESCs | B6J | 383 | 99.7% | 99.7% | 0.0% |
| 003-Clone HBG3 mESCs | B6J | 377 | 98.2% | 62.5% | 10.9% |

**Appendix 1—table 2.** Summary of data from mice treated with FK506, including graft outcome (where applicable).

List of experimental mice, including genotype, sex, age at start and end of experiment, initial and final body mass and % change (values that decreased or failed to increase are highlighted in red). Engrafted ChR2⁺ motor neurons were histologically determined to be present in all animals. Age at which ipsilateral hindlimb motor deficits were initially observed in vivo and post-mortem observation of intraneural tumour formation: *, **, and *** denotes small, medium and large tumour size, respectively (see *Figure 2E*). †Denotes animals that died during the course of treatment.

| ID | Genotype | Sex | Age (g) | | Body mass (g) | | δ BM (%) | Onset (days) | Tumor |
|---|---|---|---|---|---|---|---|---|---|
| | | | Start | End | Initial | Final | | | |
| 14.1 c | Wild-Type | M | 71 | 93 | 24.8 | 26.3 | 106.0 | - | * |
| 14.1 h | Wild-Type | F | 71 | 98 | 19.1 | 20.5 | 107.3 | 82 | - |
| 14.1 a | SOD1$^{G93A}$ | M | 71 | 100 | 26.5 | 25.5 | 96.2 | 82 | ** |
| 14.1b | SOD1$^{G93A}$ | M | 71 | 101 | 25.6 | 26 | 101.6 | 82 | ** |
| 14.1 g | SOD1$^{G93A}$ | F | 71 | 98 | 19.4 | 19.1 | 98.5 | 80 | * |
| 14B.1a | SOD1$^{G93A}$ | M | 57 | 83 | 25.2 | 21 | 83.3 | 69 | *** |
| 14B.1b | SOD1$^{G93A}$ | M | 57 | 101 | 26.3 | 27.2 | 103.4 | 70 | ** |
| 14B.1c | SOD1$^{G93A}$ | M | 92 | 121 | 28.2 | 28.4 | 100.7 | 101 | not recorded |
| 14B.2a | SOD1$^{G93A}$ | F | 90 | 124 | 21.1 | 22.9 | 108.5 | 104 | not recorded |
| 14B.2b | SOD1$^{G93A}$ | F | 90 | 124 | 20.8 | 22.3 | 107.2 | 104 | not recorded |
| 14B.2c | SOD1$^{G93A}$ | F | 90 | 127 | 20.2 | 22.2 | 109.9 | 104 | not recorded |
| 14B.2d | SOD1$^{G93A}$ | F | 90 | 120 | 21.9 | 22.2 | 101.4 | 104 | not recorded |
| 14B.2e | SOD1$^{G93A}$ | F | 90 | 117 | 22.4 | 20.7 | 92.4 | 104 | *** |
| 13B.1a | SOD1$^{G93A}$ | M | 90 | 113 | 24.2 | 21.1 | 87.2 | 101 | ** |
| 13B.2b | SOD1$^{G93A}$ | M | 85 | 112 | 23.8 | 23.5 | 98.7 | 101 | ** |

Animals listed below underwent FK506 treatment in the absence of intraneural engraftment

| ID | Genotype | Sex | Start | End | Initial | Final | δ BM (%) | Onset (days) | Tumor |
|---|---|---|---|---|---|---|---|---|---|
| 20.1d | Wild-Type | F | 101 | 131 | 19.2 | 20.95 | 109.1 | Not applicable | |
| 20.1 f | Wild-Type | F | 101 | 131 | 21.3 | 23.3 | 109.4 | | |
| 20.1 h | Wild-Type | F | 101 | 131 | 18.1 | 21.2 | 117.1 | | |
| 33.1e | Wild-Type | F | 95 | 128 | 21.1 | 22.5 | 106.6 | | |
| 33.1 f | Wild-Type | F | 95 | 128 | 22 | 22.5 | 102.3 | | |
| 33.1 g | Wild-Type | F | 95 | 128 | 21.3 | 21 | 98.6 | | |
| 33.1b | Wild-Type | M | 95 | 107† | 26.7 | 24 | 89.9 | | |
| 20.1e | SOD1$^{G93A}$ | F | 101 | 131 | 17.5 | 19.8 | 113.1 | | |
| 20.1 g | SOD1$^{G93A}$ | F | 101 | 131 | 17 | 21.2 | 124.7 | | |
| 33.1 a | SOD1$^{G93A}$ | M | 95 | 111† | 25.8 | 23 | 89.1 | | |
| 33.1 c | SOD1$^{G93A}$ | M | 95 | 128 | 24.7 | 25.8 | 104.5 | | |
| 33.1d | SOD1$^{G93A}$ | M | 95 | 128 | 23.2 | 23.2 | 100.0 | | |

**Appendix 1—table 3.** Summary of data following in vivo engraftment of ChR2[+] motor neurons in SOD1G93A mice.

Table shows a full list of all experimental animals reported in the optical stimulation section of this study, including animal ID, sex, age at start and end of study period, duration of graft, body mass at the start and end of the study period and change in body mass. The table shows the three main cohorts reported in this study, based on type of motor neurons that were engrafted and presence/absence of optical stimulation training. All animals exhibited a positive graft survival, determined by histology and/or acute optical nerve stimulation (ONS) at the experimental end-point; data is shown for 20Hz ONS, which elicited maximal tetanic contractile force.

| 2*ID | 2*Sex | Age (g) | | 2*Duration (d) | BM (g) | | 2*Δ BM (%) | 2*Max Force (g) |
|---|---|---|---|---|---|---|---|---|
| | | Start | End | | Start | End | | |
| Fast-firing Motor neurons (derived from 7DD dissociated EBs pretreated with MMC) | | | | | | | | |
| 21.1 a | M | 103 | 139 | 36 | 25.2 | 23 | 91.3 | 0.39 |
| 21.1e | M | 103 | 140 | 37 | 23.3 | 21.7 | 93.1 | 0.33 |
| 25A1b | M | 118 | 137 | 19 | 28.3 | 23.2 | 82.0 | 0.55 |
| 25 A.1d | M | 118 | 137 | 19 | 27 | 23.2 | 85.9 | 0.54 |
| 24.2 a | M | 117 | 137 | 20 | 27.2 | 23.6 | 86.8 | 0.57 |
| 25B.1b | M | 109 | 135 | 26 | 24.2 | 21 | 86.8 | 0.47 |
| 25 C.1a | M | 101 | 137 | 36 | 26.3 | 24.5 | 93.2 | 0.88 |
| 21.1 f | F | 103 | 140 | 37 | 20.8 | 19 | 91.3 | 0.43 |
| 22.1d | F | 100 | 120 | 20 | 20.5 | 19.5 | 95.1 | 1.10 |
| 22.1e | F | 100 | 121 | 21 | 23.4 | 20.4 | 87.2 | - |
| 24.2d | F | 114 | 137 | 23 | 19.5 | 18.2 | 93.3 | 0.50 |
| 25B.1c | F | 109 | 135 | 26 | 19.8 | 19.2 | 97.0 | 1.04 |
| 25 C.1d | F | 100 | 122 | 22 | 18.5 | 18.6 | 100.5 | 1.11 |
| 25 C.1f | F | 100 | 138 | 38 | 20.2 | 18.5 | 91.6 | 1.48 |
| Average: | | 106.8±7.2 | 133.9±7.2 | 27.1±7.8 | 23.2±0.7 | 21.0±0.9 | 91.1±1.6 | 0.72±0.1 |
| Slow-firing Motor neurons (derived from 5DD dissociated EBs pretreated with MMC) - untrained | | | | | | | | |
| 28.2d | F | 97 | 142 | 45 | 20.3 | 18.4 | 90.6 | 0.57 |
| 28.2 f | F | 97 | 142 | 45 | 20.2 | 17.8 | 88.1 | 0.63 |
| 28.1b | F | 104 | 136 | 32 | 22 | 19.3 | 87.7 | 0.33 |
| 26.1 a | M | 104 | 143 | 39 | 24.8 | 22.3 | 89.9 | 0.24 |
| 28.2b | M | 97 | 128 | 31 | 26.2 | 23.1 | 88.2 | 1.26 |
| 31.1 c | M | 95 | 137 | 42 | 28 | 23.9 | 85.4 | 0.64 |
| 1.1 a | M | 90 | 125 | 35 | 25.9 | 26 | 100.4 | 1.31 |
| 1.1b | M | 90 | 126 | 36 | 28.9 | 30.1 | 104.2 | 2.24 |
| 35–1 c | M | 93 | 124 | 31 | 26.4 | 25.9 | 98.1 | 0.22 |
| 35-1b | M | 93 | 130 | 37 | 29 | 28.3 | 97.6 | 0.97 |
| 35-1e | M | 93 | 130 | 37 | 23.7 | 24 | 101.3 | 0.46 |
| 40–1 c | M | 95 | 133 | 38 | 24 | 23 | 95.8 | - |
| Average: | | 95.7±4.6 | 133±6.9 | 37.3±4.8 | 25.0±0.7 | 23.5±0.9 | 93.9±1.7 | 0.81±0.18 |
| Slow-firing Motor neurons (derived from 5DD dissociated EBs pretreated with MMC)+optical stimulation training | | | | | | | | |
| 28.2 c | M | 97 | 128 | 31 | 25.6 | 24.4 | 95.3 | 8.96 |

*Appendix 1—table 3 Continued on next page*

*Appendix 1—table 3 Continued*

| 2*ID | 2*Sex | Age (g) | | 2*Duration (d) | BM (g) | | 2*Δ BM (%) | 2*Max Force (g) |
|------|-------|---------|-----|---------------|--------|-----|-----------|-----------------|
|      |       | Start   | End |               | Start  | End |           |                 |
| 1.1 c | M | 90 | 121 | 31 | 23.4 | 22.6 | 96.6 | 6.81 |
| 35-1d | M | 93 | 131 | 38 | 27.4 | 28 | 102.2 | 8.69 |
| 40–1 a | M | 95 | 132 | 37 | 25.1 | 24.2 | 96.4 | 3.60 |
| 38–1 c | F | 93 | 136 | 43 | 20.2 | 19.8 | 98.0 | 10.67 |
| 38.1e | F | 95 | 142 | 47 | 22.4 | 19 | 84.8 | 8.73 |
| 45.1 c | F | 96 | 137 | 41 | 21.6 | 21.4 | 99.1 | 4.99 |
| Average: | | 94.1±2.3 | 132.4±6.8 | 38.3±6.0 | 23.7±1.0 | 22.8±1.2 | 96.1±2.2 | 7.49±0.94 |

