## [Editor Report · eLife assessment]

This **fundamental** study presents a valuable method to restore muscle innervations in ALS mouse models using optogenetics. It is **convincing** that embryonic stem cell derived motor neurons can be transplanted into and applied to reinnervate the muscles in an ALS mouse model. The work will be of broad interest to researchers and medical biologists to develop new strategies for the treatment of neurodegenerative disorders resulting from denervated skeletal muscles.

---

## [Referee Report · Reviewer #1 (Public Review)]

Amyotrophic lateral sclerosis (ALS) is a neurodegenerative disorder leading to the loss of innervation of skeletal muscles, caused by the dysfunction and eventual death of lower motor neurons. A variety of approaches have been taken to treat this disease. With the exception of three drugs that modestly slow progression, most therapeutics have failed to provide benefit. Replacing lost motor neurons in the spinal cord with healthy cells is plagued by a number of challenges, including the toxic environment, inhibitory cues that prevent axon outgrowth to the periphery, and proper targeting of the axons to the correct muscle groups. These challenges seem to be well beyond our current technological approaches. Avoiding these challenges altogether, Bryson et al. seek to transplant the replacement motor neurons into the peripheral nerves, closer to their targets. The current manuscript addresses some of the challenges that will need to be overcome, such as immune rejection of the allograft and optimizing maturation of the neuromuscular junction.

---

## [Referee Report · Reviewer #2 (Public Review)]

The authors provide convincing evidence that optogenetic stimulation of ChR2-expressing motor neurons implanted in muscles effectively restore innervation of severely affected skeletal muscles in the aggressive SOD1 mouse model of ALS, and concluded that this method can be applied to selectively control the function of implicated muscles, which was supported by convincing data presented in the paper.

---

## [Author Response]

The following is the authors’ response to the original reviews.

**Reviewer #1 (Recommendations For The Authors):**
Major concerns:1. In lines 41-43, there seems to be some confusion about the terminology regarding "sporadic ALS". ALS is subdivided into familial and sporadic forms. Familial ALS simply indicates that the patient has a family history of ALS and presumably has a genetic predisposition for developing this disease. In many families, the identity of the mutation remains unknown. Sporadic ALS patients do not have a family history of this disease. However, this does not imply that they lack mutations that caused disease. In fact, 5-10% of these patients have the hexanucleotide repeat expansion in C9orf72. This mutation is also found in about 40% of familial ALS cases.

We have now amended the manuscript to be more accurate in our description of underlying genetics of ALS. This changes to this section are as follows:

Lines 39-47:

"...The median survival time in ALS, from initial onset of symptoms to death, typically as a result of respiratory complications, is only 20-48 months Chiò et al. (2009) and ALS has an estimated global mortality of 30,000 patients per year Mathis et al. (2019).

ALS is typically classified into either familial (fALS) or sporadic (sALS) forms of the disease, based on whether or not patients have an identified family history of the disease; between 5-10% of total ALS cases fall into the former category, fALS, with the remaining 90-95% consisting of sALS cases Mathis et al. (2019). To date, over 20 monogenic mutations that cause ALS have been identified, however these still only account for 45% of fALS cases and only 7% of sALS cases Mejzini et al. (2019)..."

2. In Fig. 4-supplement 1, 7DD and 5DD are not defined. I assume one is the fast-firing and one is the slow-firing motor neurons. I am also a bit confused as to why the 5DD neurons produce greater muscle force than the 7DD neurons when electrically stimulated. It seems to suggest that there is some difference between the two types of neurons or the groups of mice used to test them.

We have now defined these terms and the amended figure legend now reads as follows:

"(A) Fast-firing motor neurons (produced using a 7-day differentiation protocol thus labelled as “7DD”) or slow-firing ChR2+ motor neurons (produced using a 5-day differentiation protocol thus labelled as “5DD”) were engrafted in age matched SOD1G93A mice… Our expectation was that fast-firing motor neurons, which normally innervate larger numbers (>100) of stronger fast-twitch muscle fibres per motor unit would elicit significantly greater contractile force when optically stimulated, compared to slow-firing motor neurons that innervate small numbers (<10) of weaker, slow-twitch muscle fibres per motor unit. Surprisingly, our data did not show any difference when using grafts consisting of fast-firing motor neurons, versus slow-firing motor neurons, at least in response to optical stimulation. The factors underlying this surprising result, and the apparent discrepancy between electrically-evoked muscle contractions in nerves that had bene engrafted with either fast or slow firing motor neurons, are likely to be highly complex; we hope to further explore this as part of a separate follow up study."

3. Along those lines, do these two subpopulations of motor neurons innervate the same set of muscle fibers? More generally, are certain types of muscle fibers preferentially innervated by this approach? Answering these questions could point to additional ways to enhance the effectiveness of this treatment approach. This should be discussed.

This point is partially addressed in our response to Point 2 above, but to further extrapolate: certainly, the phenotype of individual muscle fibres is largely dictated by the firing properties of the motor neuron that innervates it. Slow-twitch muscle fibres tend to produce less contractile force but are more fatigue resistant, whereas fast-twitch muscle fibres produce more force but fatigue rapidly. There is evidence that expression of the chemorepellent molecule ephrin-A3 prevents the inappropriate innervation of slow-twitch muscle fibres by fast-firing motor neurons, which express the cognate receptor EphA8 [PMID: 26644518]. Importantly, fast-firing motor neurons are preferentially susceptible to disease mechanisms in ALS and the fast-twitch muscle fibres that they innervate are therefore more likely to undergo denervation and atrophy. Surprisingly, in this study we clearly show that grafts consisting of slow-firing motor neurons are able to innervate all regions of the triceps surae muscle group, including the normally exclusively fast-twitch superficial regions of the gastrocnemius and the exclusively slow-twitch soleus muscle. This finding strongly suggests that the normal developmental pairing of motor neuron and muscle fibre properties is not essential in this therapeutic context. Indeed, the use of more disease-resistant slow-firing motor neurons may provide some advantages. Again, we hope to be able to further explore this relationship in forthcoming follow-up studies.

4. The authors state that exercise programs are likely to accelerate disease progression. This is not supported by the current body of clinical data. In fact, current guidelines are for moderate (not strenuous) exercise, and mouse studies have demonstrated a protective effect of moderate exercise on disease progression.

We apologise for the lack of clarity on this point, as it was not our intention to imply that voluntary exercise accelerates disease progression. We have now amended the manuscript to specify “ENS-based exercise programs” to avoid any confusion.

5. It is unclear what the experimental endpoint is. Page 25 defines it as 135 days of age, but ranges are given the figure legends, suggesting that some other criteria were used. It also seems unclear at what determined the age at which each animal was treated since they were also not treated at the same age.

We hope that our response in the Public Reviews section above has fully addressed this point.

6. I am a little confused by Figure 5 - figure supplement 5, panel D. Why do the authors give specific p-values here but not in the other panels? The sample sizes in D are very low, in some cases with only 1 animal in a group, and performing statistical tests under these conditions seems futile. The statistical power is nearly zero.

For the purposes of consistency, we have now replaced the specific p-values in panel D with “ns”. The low n-values for the MUNE analysis data is due to the extremely difficult nature of identifying the contribution of individual motor units to the total muscle contractile response, when the maximal muscle force is extremely weak. In the absence of optical stimulation training, the extremely weak force elicited by acute optical stimulation precluded our ability to separate out the contribution of individual motor units and, often, in animals where this was not possible, we did not always perform electrically-evoked MUNE analysis. Unfortunately, we are not currently in a position to increase the n-values for this component of the study. Our ongoing research to enhance the amplitude of the muscle response to optical stimulation will hopefully help to more clearly address this in the future.

7. One concern about this approach is that the procedure could accelerate the denervation of the target muscle. Figure 5 - figure supplement 6, panel B, indicates a significant reduction in force on the ipsilateral side relative to the contralateral side, at least under electrical stimulation of the nerve. This would be consistent with the hypothesis that the procedure does enhance disease progression in the treated limb. Is there a reduction in voluntary motor activity in these animals, such as in grip strength or the position of the foot while walking?

We hope that this important point has been satisfactorily addressed in the Public Reviews section. Unfortunately, we did not undertake any behavioural analysis relating to voluntary motor function of the engrafted (or contralateral) hindlimbs, which may have provided useful data to address this point. As described above, the most likely explanation for this finding is due to physical nerve damage caused by the intraneural injection procedure; in our efforts to refine our strategy and move it towards clinical translation, we will take this into consideration in our future research.

8. Based on Fig. 6D, it seems that the vast majority of innervated NMJs at endpoint are innervated by cells from the graft. And yet, electrical stimulation evokes substantially greater muscle force. This may suggest that optical control of engrafted motor neurons will not yield enough force for routine tasks or that the few remaining endogenous motor neurons are much more effective at generating force. These potential limitations and ways to overcome them should be discussed.

There appears to be a slight misunderstanding, since our aim here was to sample a sufficiently powered number of motor end-plates innervated by YFP+ for statistical analysis. To do this we specifically chose regions of interest containing at least 1 YFP+ NMJ and the adjacent muscle fibres were included at random, whatever their innervation status. Had we sampled regions of interest at random, we would have been likely to capture only a very few YFP+ terminal as they occupy a very small volume of the total muscle section and the maximum scanning area for each high-resolution z-confocal stack is relatively small, so we feel that this selection was warranted.

Minor comments:1. The donor mouse strain should be described as 129S1/SvImJ.

We have now corrected this.

2. The first time the supplementary figures show up in the manuscript, they seem to have two titles each, such as "Figure 1-figure supplement 1. (Figure 4 - figure supplement 1)". The second seems to be the correct one.

This was caused by an issue with the Latex template, which has now been resolved.

3. PCB is not defined the first time it is used (page 8, line 332).

We have now defined this term on first use: printed circuit board (PCB)

4. CNI is not defined in the text (page 12, line 432).

We have now defined this abbreviation at the first usage on Page 4, Line 158

5. Some of the fonts on the graphs are very small, such as Fig. 5J.

We have increased the font size as much as possible for Fig. 5.

6. Figure 6 - figure supplement 1 does not include a key to indicate which antigens are stains and which color refers to which antigen. This is also needed for the videos.

We have now included a key on this figure supplement to indicate the relevant antigens and stain and we have also done the same for the videos.

7. Video 5 seems to indicate that there is a dead zone in the back of the chamber. Does this raise any concerns about the consistency of training from animal to animal?

This is an extremely astute observation. However, the intermittent activation of the implantable LED devices is not due to a dead zone; rather, it is due to the orientation of the power receiving coil within the device and it’s alignment with the resonance frequency chamber that transmits the power to the device. As the animals move around, and particularly when they rear up, the power receiving coil occasionally becomes misaligned and fails to receive sufficient power to activate the LED. Since the pulses are delivered every 2 seconds, for 1 hour per day, we feel that the animals, on average, receive sufficient numbers of pulses to implement the training regimen. Indeed, we feel that the results speak for themselves.